# Macrocyclization of linear molecules by deep learning to facilitate macrocyclic drug candidates discovery

Yanyan Diao[1], Dandan Liu[1], Huan Ge[1], Rongrong Zhang[1], Kexin Jiang[1], Runhui Bao[1], Xiaoqian Zhu[1], Hongjie Bi[1], Wenjie Liao[1], Ziqi Chen[1], Kai Zhang[2], Rui Wang[1], Lili Zhu[1], Zhenjiang Zhao[1], Qiaoyu Hu[2] & Honglin Li [1,2,3] ✉

Interest in macrocycles as potential therapeutic agents has increased rapidly. Macrocyclization of bioactive acyclic molecules provides a potential avenue to yield novel chemical scaffolds, which can contribute to the improvement of the biological activity and physicochemical properties of these molecules. In this study, we propose a computational macrocyclization method based on Transformer architecture (which we name Macformer). Leveraging deep learning, Macformer explores the vast chemical space of macrocyclic analogues of a given acyclic molecule by adding diverse linkers compatible with the acyclic molecule. Macformer can efficiently learn the implicit relationships between acyclic and macrocyclic structures represented as SMILES strings and generate plenty of macrocycles with chemical diversity and structural novelty. In data augmentation scenarios using both internal ChEMBL and external ZINC test datasets, Macformer display excellent performance and generalisability. We showcase the utility of Macformer when combined with molecular docking simulations and wet lab based experimental validation, by applying it to the prospective design of macrocyclic JAK2 inhibitors.

Macrocycles, typically defined as cyclic small molecules or peptides with ring structures consisting of 12 or more atoms, has emerged as promising chemical scaffolds in the field of new drug discovery[1,2]. The distinct physicochemical properties, including high molecular weight and abundant hydrogen bond donors[3], render this structural class occupy a chemical space beyond Lipinski's rule of five[4]. In comparison to their linear analogs, macrocycles tend to adopt pre-organized constrained conformations and establish extended contacts with targets. Consequently, they have the potential to exhibit enhanced binding affinities, improved selectivities or superior pharmacological characteristics[5,6]. Macrocycles have been successfully used as potential therapeutic agents for various pharmaceutical targets, such as kinases, proteases and G-protein-coupled receptors. In particular, due to the distinctive features, macrocycles are

regarded as a privileged chemotype for targeting some challenging proteins that are hardly tractable by traditional small molecule drugs[7], thus bridging the gap between small molecules and large biologics. For instance, macrocycles predominate the marketed inhibitors of hepatitis C virus NS3/4A, which possesses a shallow and solvent-exposed groove that poses challenges for small molecule binding[8]. The advantages of macrocycles have also been reported in modulating protein-protein interactions with large, flat, and dynamic surfaces[9].

In addition to naturally occurring macrocycles, synthetic analogs derived from principles of medicinal chemistry are another important source of macrocyclic compounds[10,11]. Macrocyclic modification of known acyclic active compounds is a straightforward and effective strategy to obtain novel macrocyclic scaffolds bypassing intellectual

[1]Shanghai Key Laboratory of New Drug Design, School of Pharmacy, East China University of Science & Technology, Shanghai 200237, China. [2]Innovation Center for AI and Drug Discovery, East China Normal University, Shanghai 200062, China. [3]Lingang Laboratory, Shanghai 200031, China. ✉e-mail: hlli@ecust.edu.cn

property restrictions, and can achieve desired pharmacological properties[12]. For instance, Lorlatinib, a FDA-approved macrocyclic inhibitor targeting anaplastic lymphoma kinase, was derived from the acyclic Crizotinib. Lorlatinib displayed improved kinase selectivity and enhanced exposure to the central nervous system[13]. This demonstrates how macrocyclic modification of known compounds can lead to the development of new and improved drugs. Although there have been more than 80 macrocyclic drugs approved for clinical use[14], macrocycles are still sparsely exploited within drug design projects, partially owing to their synthetic intractability and deficiency of efficient macrocyclization approaches[15].

Given a biologically active linear molecule as the starting point, the reported successful rational design of macrocycles generally involves two key steps. Firstly, there is the addition of macrocyclic linkers that are compatible with the linear compound, resulting in the formation of macrocycles. Secondly, the compatibility between the macrocycles and the binding pocket of the target is evaluated. For the second step, available research methods are relatively explicit, and many simulation methods commonly used in drug design, such as conformation optimization and molecular docking, can assist this process. If we can generate abundant macrocycles with chemical diversity by adding structurally diverse linkers in the first step, the chance of obtaining novel macrocyclic candidates after subsequent target-compound binding prediction will undoubtedly increase. Nevertheless, the macrocyclization of linear compounds in the initial stage is primarily driven by the empirical knowledge of the medicinal chemists. While the final results are often presented, the detailed procedures involved are often inadequately described in scientific literature. This opaque and non-standardized procedure is difficult for inexperienced researchers to follow, and the empirical knowledge is insufficient to cover the vast chemical space of the macrocyclic linkers.

Although rarely mentioned in publications, computational tools have demonstrated successful applications in facilitating the macrocyclization process[16,17]. Wagner et al.[18] and Sindhikara et al.[19] utilized geometrically constrained linker database searching and linker connection strategy to generate macrocycles from acyclic ligands. In the studies, a conformer ensemble of linker fragments were first constructed, from which the linkers were usually filtered by applying geometric criteria, e.g., distance and angle compatibility between the atoms to be connected, to form initial macrocycles using the three-dimensional (3D) structure of the acyclic ligand. Using molecular docking, MM/GBSA, and/or free-energy perturbation calculation in combination to assess the interactions with the target, promising candidates were identified from the generated macrocycles. However, these methods can only enumerate pre-built linker libraries, without the ability to derive new structurally novel linkers. Additionally, the focus on local conformational matching of connected atoms may not provide a comprehensive understanding of the overall macrocyclic structure. Besides, these tools are not publicly available, which hinders widespread use and collaboration. As concerns on the studies of macrocyclic drug candidates have grown at a remarkable rate in both industry and academic institutions, there is an urgent need to develop practical computational tools to assist in the cyclization of linear bioactive molecules.

Artificial intelligence, particularly deep learning technology, has exhibited great potentials in various stages of drug discovery process, including de novo molecule generation, scaffold hopping, structural optimization, and activity prediction[20–24]. However, training neural networks typically requires large amounts of data in order to achieve high precision and generalization ability[25]. Hence, current applications of deep learning in the field of drug development have mainly focused on drug-like small molecules. To the best of our knowledge, the implementation of macrocyclization for linear molecules by utilizing deep learning algorithms remains an underexplored area. The underneath reasons are complicated, while the relatively small number of

macrocycles available for model training, rooted from their long-term underexploited state, is probably the most relevant one.

Chemical molecules can be represented as Simplified Molecular Input Line Entry System (SMILES) strings[26], a chemical language naturally suitable for sequence-based deep learning models. Different SMILES representations of a same chemical structure have been used as a data augmentation method to obtain generalizing models for small-data regimes[27–29]. Herein, we propose a Transformer-based model called Macformer for automated macrocyclization. Given a linear compound represented as a SMILES string with two cyclization site labels, Macformer aims to explore the vast chemical space of its macrocyclic analogs by leveraging the benefits of deep generative models. Unlike the computational cyclization methods mentioned above, Macformer tackles the macrocyclic skeleton design problem as a machine translation task through handling SMILES sequences end-to-end. By employing a data augmentation strategy with randomized SMILES strings, Macformer efficiently learns the implicit mapping relationships between the SMILES syntax of acyclic and macrocyclic structures. It can automatically fill in the missing linker of the input acyclic fragment to generate corresponding macrocyclic scaffolds with chemical diversity and structural novelty. We applied Macformer to the macrocyclization of Fedratinib, a FDA-approved JAK2 inhibitor. As macrocycles were inferred without the constraints of specific targets in Macformer, the macrocyclic analogs of Fedratinib generated by Macformer were subjected to molecular docking calculations. Based on the docking poses in the ATP binding site of JAK2 and an estimation of synthetic accessibility, three macrocycles were ultimately selected for synthesis and testing through both in vitro and in vivo experiments. The representative compound **3** has improved kinase selectivity and pharmacokinetic properties than Fedratinib. Notably, it displays comparable in vivo efficacy to Fedratinib at a lower dose. These results demonstrate the great potential of Macformer in the discovery of macrocyclic drug candidates.

## Results
### Model overview
Starting from an acyclic bioactive molecule, sampling broad chemical space of macrocyclic linkers would efficiently improve the hit probability of macrocyclic lead compounds. The schematic representation of the Macformer framework is illustrated in Fig. 1. It is a deep generative model designed to generate diverse and novel macrocyclic analogs of the given acyclic molecules. Due to the absence of explicit targets for many bioactive macrocycles, the target information is not involved in Macformer.

We collected 18357 bioactive macrocycles from ChEMBL database[30], and the filter conditions are that the number of macro ring with 12 or more atoms is inferior to 1 and the SMILES strings length is inferior to 200. To mimic the real macrocyclization process, through traversing every combination of two single bonds on the macro ring and subsequent linker filtration, 237728 unique acyclic-macrocyclic SMILES pairs were yielded for model training and evaluation (see more details in Data Preparation section of Methods part). The data processing procedure can be regarded as the reverse process of macrocyclization (Fig. 1a) which dramatically increased the number of data available for deep learning. The acyclic SMILES strings, containing dummy atoms (*) to label the cyclization site, represent the linear compounds to be macrocyclized and will be fed to the neural network as source sequences. The macrocyclic SMILES strings are the target sequences expected to be output by the model. Consequently, the macrocyclization problem is tailored as a chemical language-based sentence completion task, where the missing linkers of the input acyclic compounds are added and the intact macrocyclic compounds are generated. Our method proposed here is based on the Transformer architecture[31], which is the state-of-the-art neural network model to deal with sequential data. Different from previously popular recurrent

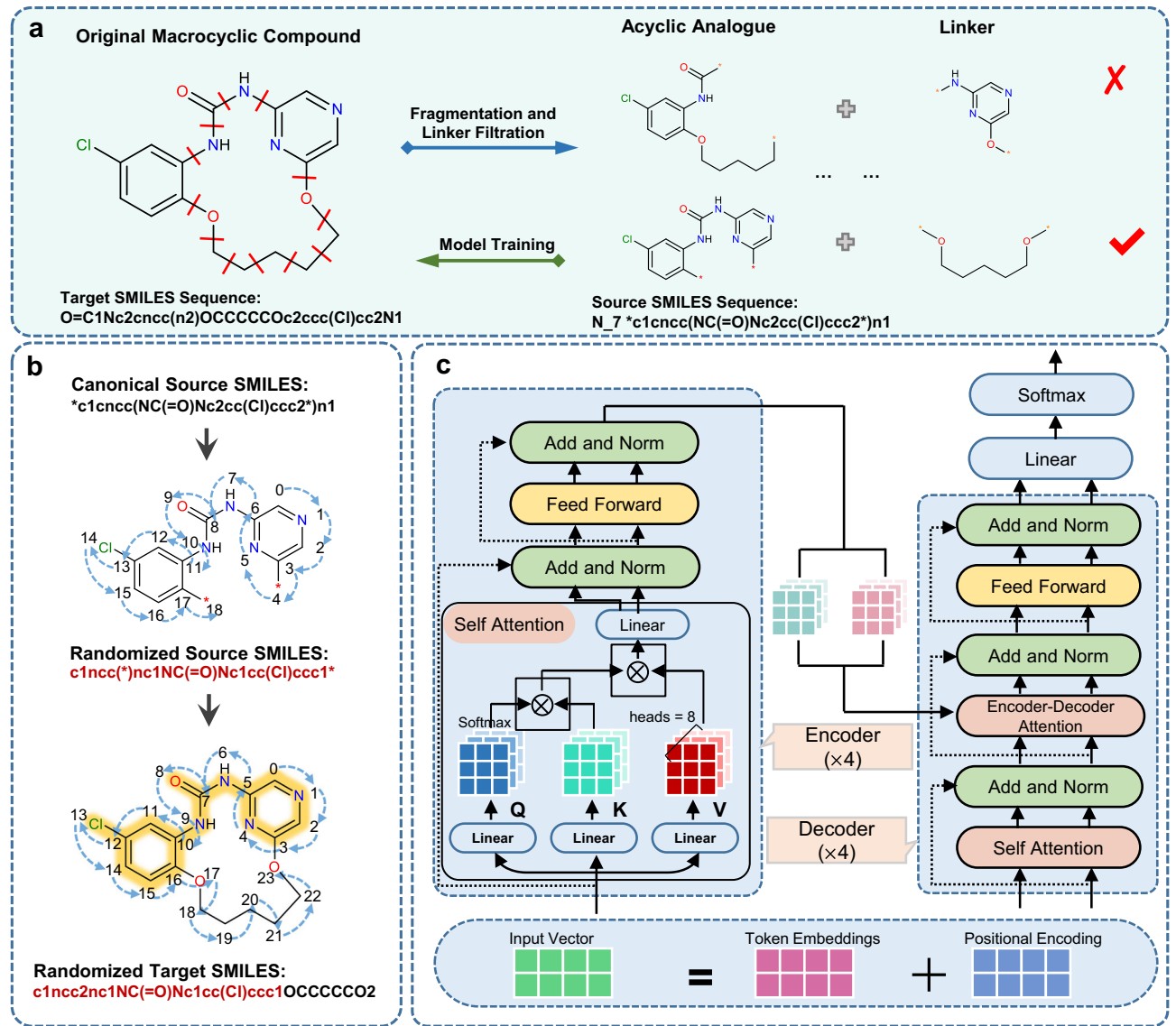

**Fig. 1 | Overview of the workflow of Macformer. a** Data preprocessing protocol to generate acyclic-macrocyclic SMILES pairs for model training and evaluation, and the "N_7" token indicates the number of heavy atoms on the shortest path of the linker. **b** Augmentation of acyclic-macrocyclic pairs using randomized SMILES in a substructure-aligned manner. **c** The model network architecture of Macformer. The scaled-dot attention layer takes three matrices as input: the matrix **Q** packed with a set of queries, the matrix **K** with keys, and the matrix **V** with values.

neural networks that process data sequentially in a token by token manner, Transformer adopts attention mechanisms and positional embeddings for holistic processing of input sequential data (Fig. 1c). The attention mechanism enables the model to capture contextual information for tokens at any position, allowing for the identification of long-range dependencies between tokens in a sequence. This means that all tokens in the source acyclic SMILES strings, albeit with varing attention weights, contribute to the generation of the macrocyclic SMILES strings in Macformer. Benefiting from the global information modeling method over the input acyclic SMILES sequences, Macformer is anticipated to infer more suitable macrocyclic linkers compatible with the given acyclic molecules and generate novel macrocycles closer to the chemical space of the bioactive macrocycles used as the training dataset. The details of our model are fully described in Macformer section of Methods part.

Canonical SMILES notation, a string representation unique to each molecule, is widely used due to its simplicity. However, recent studies have shown that data augmentation by using a batch of chemically identical but syntactically different randomized SMILES during training and inference can greatly modify the performance of deep learning methods[28,32]. In order to improve the quality of our model, data augmentation was performed for both source and target sequences of the training dataset. Notably, the input acyclic scaffolds are substructures of the output macrocycles. If we feed this prior knowledge to the model in the form of aligned SMILES strings, it will help the model understand the relationship between input and output sequences. During this process, randomized SMILES of the acyclic scaffolds were firstly generated by randomly selecting the starting atom and the direction of the molecular graph enumeration. Subsequently, substructure searching were performed and the atom numbers of the macrocycles were re-ordered according to that of the acyclic substructure. The randomized SMILES of the macrocycles were finally acquired based on the new atom numbers (Fig. 1b). Such a substructure-aligned strategy minimizes the gap between the input and output sequences, favoring the model to pay more attention to the inference of macrocyclic linker.

Four models with different augmentation levels (none, ×2, ×5, and ×10) were trained. The non-augmented scenario contains only

**Tabel 1 | Comparison of Macformer with different augmentation numbers and MacLS on ChEMBL test dataset**

| Method | aug.[a] | Recovery (%) | Validity (%) | Uniqueness (%) | Novelty_mol (%) | Novelty_linker (%) | Macrocyclization (%) |
|---|---|---|---|---|---|---|---|
| Macformer[b] | None | 54.85 ± 14.28 | 66.74 ± 2.29 | 63.18 ± 6.38 | 89.30 ± 1.94 | 40.56 ± 2.33 | 95.00 ± 0.74 |
| | ×2 | 96.09 ± 0.61 | 80.34 ± 1.38 | 64.43 ± 0.23 | 91.58 ± 0.15 | 58.91 ± 0.36 | 98.62 ± 0.17 |
| | ×5 | 97.54 ± 0.16 | 81.94 ± 1.42 | 65.36 ± 0.13 | 91.79 ± 0.16 | 62.11 ± 0.65 | 98.80 ± 0.11 |
| | ×10 | 97.02 ± 0.05 | 82.59 ± 1.57 | 64.44 ± 0.46 | 91.76 ± 0.22 | 60.27 ± 0.96 | 98.46 ± 0.04 |
| MacLS_self[c] | / | 0.01 ± 0.01 | 17.05 ± 0.29 | 95.33 ± 0.01 | 100 ± 0.00 | 0.00 ± 0.00 | 100 ± 0.00 |
| MacLS_extra[c] | / | 4.16 ± 0.20 | 89.65 ± 0.03 | 96.32 ± 0.06 | 99.65 ± 0.02 | 0.00 ± 0.00 | 100 ± 0.00 |

[a]The fold of augmentation of ChEBML training dataset.

[b] Data are mean ± SD, $n$ = 10 independent experiments using different source SMILES strings. Source data are provided as a Source Data file.

[c] Data are mean ± SD, $n$ = 3 independent experiments using top 3 low-energy conformations. Source data are provided as a Source Data file.

canonical SMILES strings. In addition to one copy of canonical SMILES, the $n$-fold augmented scenarios contain $n−1$ chemically equivalent but randomized SMILES. All models converge very well after running for 50,000 steps (Supplementary Fig. 1). During the evaluation process of the model using test datasets, each experiment was performed ten times using various SMILES strings. This approach allows us to assess the predictive ability of the trained models across diverse SMILES representations. Beam search algorithm[33] was applied to infer multiple candidate sequences on the test datasets, and top 10 predictions were generated for each input sequence.

To compare our model with previously reported non-deep learning computational approaches for automatic macrocyclization, we propose a pipeline to construct macrocycles from 3D structures of linear compounds through linker database searching (termed as MacLS, Supplementary Fig. 2), following the works of refs. 18,19. In MacLS, the linkers were selected based on the compatibility of attachment vectors between the acyclic compound and the linker. For an acyclic molecule or a linker, an attachment vector is the bond between the atom at the cyclization site and the leaving atom that will not be included in the generated macrocycles. Likewise, the target information is not considered in MacLS for a fair comparison to Macformer. The linkers of the ChEMBL training dataset were used and a conformer ensemble containing 163,924 structures was produced. For internal ChEMBL and external ZINC test datasets, the conformations of the linear chemical structures were obtained in two ways. The first approach involves generating conformations from scratch based on the SMILES strings of the acyclic structures (termed as MacLS_self). The second approach involves extracting the conformations from the low-energy 3D structures of corresponding target macrocycles (termed as MacLS_extra). The linker database was first enumerated and filtered through the distance and dihedral angle constraints of atoms on the attachment vectors. By connecting the remained linkers to the acyclic skeleton, the macrocyclic compounds were acquired, ranked by the root-mean-square deviation (RMSD) values between the atoms of the attachment vectors in the acyclic fragments and that in the linkers. Top 10 macrocycles were reserved for each acyclic structure in internal ChEMBL and external ZINC test datasets for a fair comparison.

### Evaluation on internal ChEMBL test dataset

Given an acyclic molecule, the purposes of Macformer and MacLS are to generate diverse and novel macrocyclic analogs before further evaluation of binding potential against the target of interest. For the specific task, the assessment criteria were not clearly indicated in previously studies. In this work, we applied the widely used metrics from deep generative models to evaluate the performance of our method. These metrics include the reconstruction of target molecule and the assessment of chemical validity, novelty, and uniqueness of the generated compounds. The novelty of infered linkers and the

macrocyclization ratio of the generated compounds were also calculated as additional metrics (see more details in Model Evaluation Metrics section of Methods part).

The performances of Macformer and the non-deep learning approach MacLS on ChEMBL test dataset are summarized in Table 1. In comparison to the baseline model without augmentation, amplifying the training dataset by two-fold brought about better performance in terms of all metrics, especially for recovery (96.09% vs 54.85%), validity (80.34% vs 66.74%), and linker novelty (Novelty_linker, 58.91% vs 40.56%). This indicates that the model trained with substructure-aligned randomized SMILES is advantageous not only for reconstructing macrocyclic skeletons but also for learning the fundamental syntax of the chemical language. Consequently, the model is capable of generating chemically meaningful SMILES strings with structurally diversity and novelty. However, models trained with five- or ten-fold data augmentations did not result in further significant improvement of performance on ChEMBL test dataset. This phenomenon is accordant to the conclusion of a previous study that an optimal degree of data augmentation is important for a given learning task[27]. It is worth noting that all models could achieve over 95% macrocyclization ratio, illustrating the ability of Macformer for the purpose of generating macrocyclic compounds. The overall uniqueness values are lower than 66%, which may be attributed to the redundancy of target compounds in the ChEMBL test dataset. Among the 23771 acyclic-macrocyclic SMILES pairs in the test dataset, there are 10,222 unique macrocycles, resulting in an external uniqueness rate of 43%. In spite of this, the macrocyclic compounds generated by Macformer exhibit significantly higher uniqueness compared to the original test dataset. The results confirm the capability of Macformer in creating diverse and previously unseen macrocyclic structures that go beyond the available ChEMBL macrocycle dataset.

For assessment of the MacLS method, the conformations of the acyclic compounds were first constructed directly from their SMILES notations. In this scenario, MacLS_self only generate 17.05% valid macrocycles. The very low validity is principally attributed to the linear extended conformations of the parent acyclic compounds that are not suitable for macrocyclization. When using the more folded conformations extracted from the preformed 3D structures of the target macrocycles, the validity of macrocycles generated by MacLS_extra is greatly improved, implying the high dependence of the non-deep learning macrocyclization method on the given conformations of the acyclic scaffold. Compared to Macformer, MacLS performs better in terms of uniqueness and molecular novelty (Novelty_mol). Nevertheless, MacLS can not derive new linkers of structural novelty, leading to the Novelty_linker values of 0%. Additionally, MacLS reconstructs the target macrocycles at very low ratios, only 0% and 4.16% for MacLS_self and MacLS_extra, respectively. The results are not surprising, as MacLS merely takes into account the geometrical constraints associated with attachment vectors, which are local information of little aid in reconstructing the target macrocycles.

**Tabel 2 | Comparison of Macformer with different augmentation numbers and MacLS on ZINC test dataset**

| Method | aug.[a] | Recovery (%) | Validity (%) | Uniqueness (%) | Novelty$_{mol}$ (%) | Novelty$_{linker}$ (%) | Macrocyclization (%) |
|---|---|---|---|---|---|---|---|
| Macformer[b] | None | 2.70 ± 1.31 | 72.91 ± 2.05 | 47.74 ± 8.98 | 96.10 ± 0.81 | 44.24 ± 2.05 | 96.39 ± 0.71 |
| | ×2 | 76.37 ± 3.23 | 81.97 ± 1.20 | 44.99 ± 5.37 | 99.31 ± 0.19 | 53.03 ± 0.65 | 99.48 ± 0.08 |
| | ×5 | 81.86 ± 0.75 | 84.73 ± 1.01 | 45.14 ± 4.60 | 99.39 ± 0.09 | 53.98 ± 1.00 | 99.53 ± 0.05 |
| | ×10 | 84.25 ± 0.84 | 85.35 ± 1.33 | 45.26 ± 0.46 | 99.43 ± 0.09 | 50.00 ± 0.95 | 99.27 ± 0.07 |
| MacLS_self[c] | / | 0.00 ± 0.00 | 13.02 ± 0.79 | 83.68 ± 0.74 | 100 ± 0.00 | 0.00 ± 0.00 | 100 ± 0.00 |
| MacLS_extra[c] | / | 4.52 ± 0.20 | 89.67 ± 0.07 | 95.04 ± 0.14 | 99.99 ± 0.00 | 0.00 ± 0.00 | 100 ± 0.00 |

[a] The fold of augmentation of ChEBML training dataset.
[b] Data are mean ± SD, $n$ = 10 independent experiments using different source SMILES strings. Source data are provided as a Source Data file.
[c] Data are mean ± SD, $n$ = 3 independent experiments using top 3 low-energy conformations. Source data are provided as a Source Data file.

## Evaluation on external ZINC test dataset

These methods were further evaluated on an additional external test dataset containing 5551 acyclic-macrocyclic SMILES pairs, which were extracted from 486 bioactive macrocycles in ZINC database. Comparing with those from ChEMBL database, these macrocycles have lower molecular weights and shorter SMILES lengths (Supplementary Fig. 3). As shown in Table 2, the augmented models can also provide systematically improved performance on the external ZINC dataset. Both models trained with 5- and 10-fold augmentations could recover over 80% of the original macrocyclic compounds, generate over 84% valid SMILES strings, and achieve over 99% novelty and macrocyclization. These results indicate that Macformer has excellent generalization ability in data augmentation scenarios.

The performance of MacLS on the external ZINC test dataset is similar to that on the ChEMBL test dataset. Fundamentally, MacLS could not learn sufficient prior knowledge through the training process like Macformer, hence the evaluation results of MacLS on the two test datasets are similar and can theoretically be extrapolated to other datasets.

## Properties of generated novel macrocycles

Irrespective of the derivation of structurally novel linkers, both Macformer and MacLS demonstrate the ability to generate macrocyclic compounds with structural novelty. This raises the question of whether there are distinctions between the chemical spaces of these novel compounds. To explore this question, we first assessed the structural similarity between generated novel and ground-truth target macrocycles using Morgan fingerprints (2 bond radius) implemented in RDKit v2020.03.3.0[34]. For a given target compound, its Tanimoto coefficient (Tc) values with all the corresponding generated novel compounds were calculated and averaged to obtain the final score. As illustrated in Fig. 2a, the majority of generated novel compounds have average Tc scores higher than 0.7, due to the common substructures between the acyclic and macrocyclic compounds. However, Macformer tend to generate new chemicals with higher structural similarity to the target macrocyclic compounds than MacLS_extra.

The above result is somewhat unexpected, since Macformer can infer novel linkers that are not present in the training dataset, whereas MacLS_extra does not possess this ability. Subsequently, we probed the chemical space of the novel linkers by calculating their 1024-bit Morgan fingerprints. Additionally, we utilized the uniform manifold approximation and projection (UMAP) algorithm[35] for dimensionality reduction. UMAP can better preserve the similarity relations between data points in the original high-dimensional space than $t$-distributed stochastic neighbor embedding[36]. As shown in Fig. 2b, the structurally novel linkers generated by Macformer on ChEMBL test and ZINC datasets are both located in the chemical space surrounding the linkers from the ChEMBL training dataset. Meanwhile, in addition to executing macrocyclization, Macformer can simultaneously introduce minor modifications on the starting

linear substructure to generate new structures with high similarity to the target macrocycles (Fig. 2c).

Furthermore, we utilized Pipeline Pilot v2017[37] to calculate seven molecular properties: molecular weight (MW), AlogP, polar surface area (PSA), the number of hydrogen bond acceptors (NHA), the number of hydrogen bond donors (NHD), quantitative estimates of drug-likeness (QED), and synthetic accessibility (SA). For both ChEMBL test and ZINC datasets, the novel macrocycles generated by MacLS_extra tend to have more significant statistical differences to the targets than those generated by Macformer (Supplementary Fig. 4 and 5). The results indicate that the chemical space of the Macformer generated novel macrocycles is closer to that of the real bioactive ones. This may benefit from the data augmentation strategy by using substructure-aligned randomized SMILES, which exposes the structural information of a same molecule from various views and renders Macformer powerful in understanding the constraints of the macrocyclic chemical space.

## Model interpretability via attention weights analysis

To disclose how Macformer works in this specific automatic macrocyclization task, attention weights between input and output sequences were analyzed from the substring and token scale, respectively (Supplementary Fig. 6). The substrings or tokens in the input sequences tend to have the greatest impact on the generation of the same substrings or tokens in the predicted sequence, which guarantee the reproduction of the starting acyclic fragment in the generated macrocycles. When inferring the macrocyclic linker substring, our model displayed a systematic manner, as the discrepancies in terms of attention weights among different substrings of the source sequence are not significant. This indicates that Macformer is able to combine the latent features of the input acyclic SMILES sequence, and incorporate appropriate linker to the original linear fragment. This capability stems from the prior knowledge it has learned about the relationship between the acyclic fragments and their corresponding macrocyclic linkers in the training dataset.

## Design of macrocyclic JAK2 inhibitors using Macformer and molecular docking

In recent years, macrocycles have gained significant attention for their potential as kinase inhibitors. For prospective evaluation purpose, Macformer was employed to design macrocyclic Janus kinase 2 (JAK2) inhibitors. JAK2 belongs to the intracellular non-receptor protein tyrosine JAK family kinases and is an important target for the treatment of myeloproliferative neoplasms and rheumatoid arthritis[38,39]. In combination of molecular docking simulation and medicinal chemistry-based analysis, William et al. designed the macrocyclic JAK2 inhibitor Pacritinib[40]. This inhibitor was derived from the heavily patented phenylaminopyrimidine structure and has been approved to treat myelofibrosis[41].

In our study, the starting acyclic structure was derived from Fedratinib, a small molecule JAK2 inhibitor approved for the

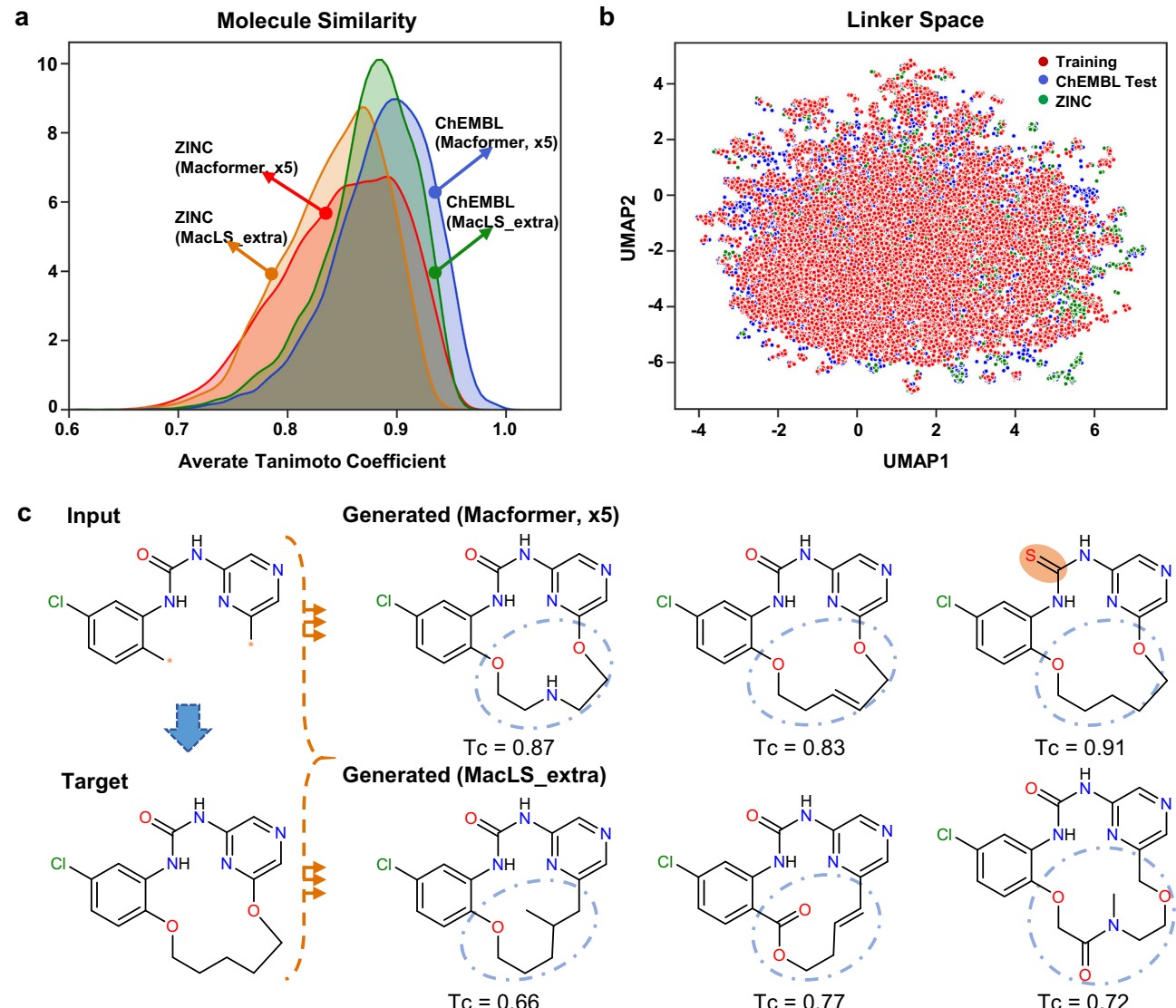

**Fig. 2 | Comparison of chemical space between novel macrocycles generated by Macformer trained with five-fold data augmentation and MacLS_extra, respectively, on ChEMBL test and ZINC datasets. a** Distribution of average Tanimoto coefficient between generated novel and ground-truth target macrocycles. ChEMBL, Macformer, ×5, $n = 23772$; ChEMBL, MacLS_extra, $n = 23765$; ZINC, Macformer, ×5, $n = 5514$; ZINC, MacLS_extra, $n = 5551$. **b** UMAP plot of the 1024-bit Morgan fingerprints of the linkers in the ChEMBL training dataset ($n = 9243$ linkers) and the novel linkers generated by Macformer on ChEMBL test ($n = 9039$ linkers) and ZINC ($n = 2082$ linkers) datasets, respectively. **c** Retrospective macrocyclization of a Checkpoint Kinase 1 (CHK1) inhibitor[64] by Macformer. The Tc values between the generated novel and target compounds were labeled. Source data are provided as a Source Data file.

therapy of myelofibrosis[42]. Fedratinib is reported to be highly selective to JAK2 over other JAK kinase, but its selective profile against the wider kinome is disappointing[43,44]. The off target effects on other kinases may cause undesirable adverse reactions. We hope that macrocyclization would obtain proprietary skeletons with improved kinase selectivity and other properties. The macrocyclization connection points were set on the two terminal phenyl rings. Meanwhile, the *tert*-butyl sulfonamide moiety, which may cause unfavorable contacts with Asp994[45], was removed in order to improve synthetic feasibility of generated macrocycles. Tokens representing the number of heavy atoms on the shortest path of the linkers ranging from three to nine were added prepended to the SMILES sequence to maximize the variety of the macrocyclic linkers. To increase the diversity of predicted macrocycles, each source SMILES sequence was augmented by ten-fold. Inferring by Macformer with beam size of 10, 700 output SMILES sequences were finally obtained, which included 281 unique novel macrocyclic molecules (Fig. 3).

The traditional MacLS method was also evaluated for its potential in macrocyclization of Fedratinib, and top 300 macrocyclic analogs were reserved based on the crystallographic bioactive conformation of Fedratinib in complex with JAK2 kinase domain (PDB code 6VNE)[45]. Using the GlideSP protocol of Maestro v10.1[46], the macrocycles generated by both methods were docked, respectively, into the ATP binding site of JAK2. The docking scores, the lower (more negative) the better, were used as the evaluation metrics. A comparison of the docking scores (Supplementary Fig. 7) showes that macrocycles generated by Macformer have lower values than those by MacLS, meaning that Macformer is more likely to generate active JAK2 macrocyclic inhibitors. We may explain this phenomenon from two aspects. On the one hand, the non-deep learning MacLS method merely consider the matching of geometric parameters related to the formation of new macrocyclic chemical bonds, in order to maintain the bioactive orientation of the linear compound. However, during the overall structural optimization or target-induced binding of the macrocyclic compounds, the conformations of many acyclic substructures may

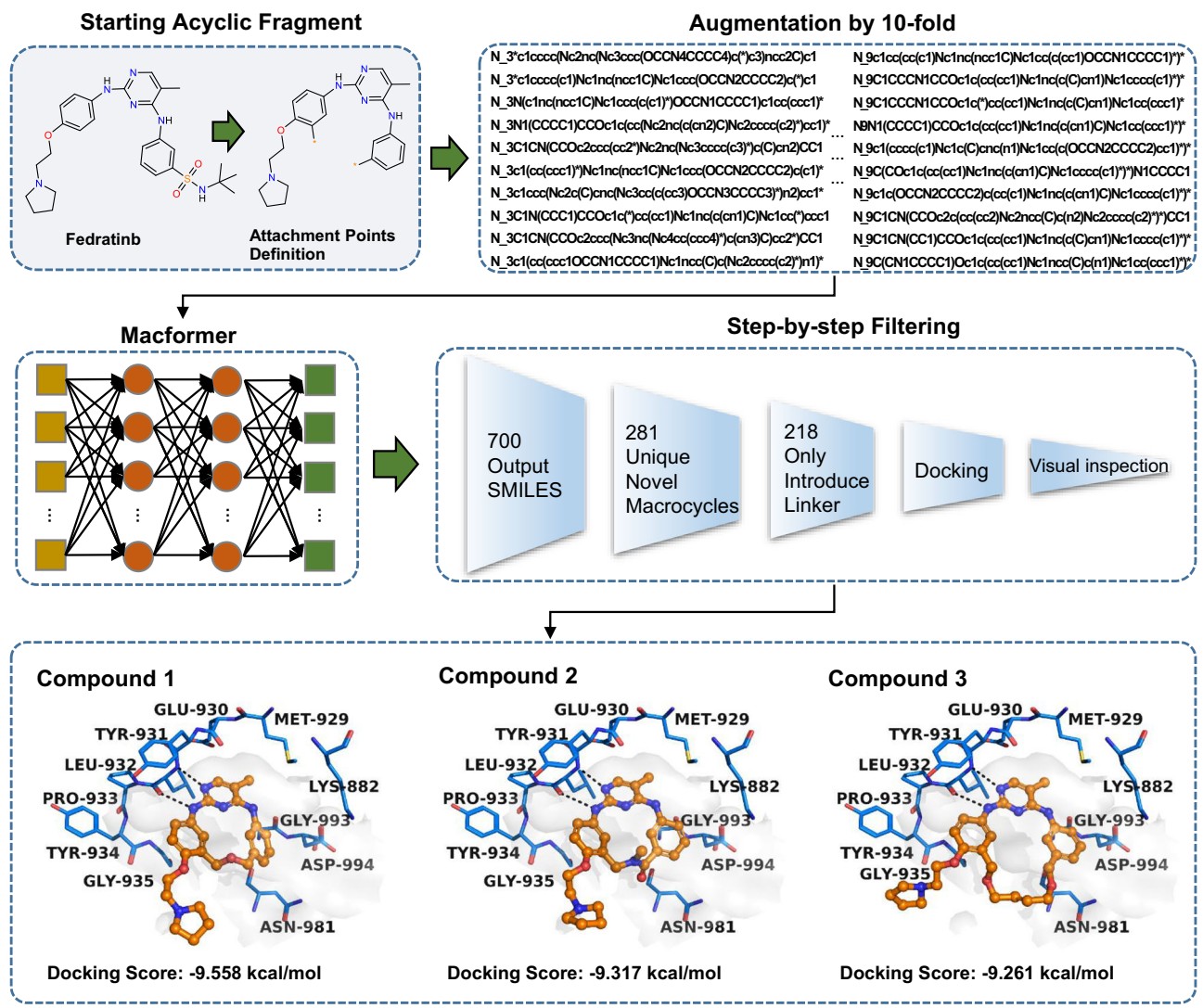

**Fig. 3 | Schematic representation of the procedure for design of macrocyclic JAK2 inhibitors starting from Fedratinib.** The macrocyclization connection points, labeled by asterisks (*) for clear identification, are deliberately located on the two terminal phenyl rings of Fedratinib. To enhance the diversity of predicted macrocycles, tokens denoting the number of heavy atoms on the shortest path of the linkers (N_x) ranging from three to nine are inserted at the beginning of the SMILES sequence. Subsequently, each source SMILES sequence is augmented by ten-fold. In the putative docking poses of the macrocyclic JAK2 inhibitors, the critical hydrogen bonds are highlighted as black dashes.

slightly change to deviate from the active one. On the other hand, Macformer could grasp the complete structural information of the acyclic and macrocyclic compounds through global modeling on SMILES sequences with Transformer architecture and infer linkers matching the given acyclic molecules. The overall compatibility of the starting acyclic compounds with the linkers may help preserve the active conformations of the acyclic substructures and facilitate the binding of the macrocycles with the target.

In the prospective case study, our efforts mainly focus on the introduction of macrocyclic linkers, hence the generated compounds with extra substituent groups on the starting fragment and those undergo scaffold hopping were ignored. The ultimately eligible 218 macrocyclic compounds generated by Macformer were reserved for further in-depth study. After visual inspection of docking poses and estimation of synthetic accessibility based on experiences, three compounds were finally selected for synthesis and evaluation for their potency against JAK2. As shown in Fig. 3, the starting acyclic moiety display similar binding pose to that of Fedratinib, and macrocyclization retains the critical hydrogen bond interactions between compounds 1–3 and residue Leu932 in the hinge region. Despite the

structural novelty of the three selected compounds, their introduced linkers are seen in many macrocycles of the ChEMBL training dataset. Specifically, the linkers of compounds **2** and **3** are present in the approved drugs Lorlatinib and Pacritinib, respectively. However, to the best of our knowledge, the linker of compound **1** has not been reported for the design of macrocyclic and selective JAK2 inhibitor. Rather than extremely pursuing the novel linkers, synthesizing these macrocyclic compounds is our main concern, which is also the premise of further activity evaluation. The presence of these linkers in authentic macrocycles generally implies better synthetic feasibility, and the three macrocycles covering three distinct chemical skeletons were ultimately synthesized through three different routes (Supplementary Methods, Supplementary Fig. 9–17).

Upon retrospective examination of the 300 macrocycles generated by MacLS, none of the three compounds were found, demonstrating the practicality of our deep learning method in identifying potent macrocyclic JAK2 inhibitors that may have been overlooked by traditional methods. Interestingly, there are only two macrocycles in common between the compounds generated by Macformer and by MacLS. Although not practiced in this study, we believe that the

chances of obtaining potent macrocyclic lead compounds will be greatly increased if compounds obtained by the two methods are combined for further investigation.

## Activities of compounds 1–3 at enzymatic and cellular level
Enzymatic assays of compounds 1–3 against JAK2 were subsequently performed and the $IC_{50}$ values of compounds 1–3 were measured to be 0.07, 0.364, and 0.006 μM, respectively (Table 3). The most potent compound 3 exhibits similar single-digit nanomolar range activity compared to Fedratinib. To assess the specificity of the two most potent macrocycles, 1 and 3, their kinase selectivity profiles against a diverse panel of 468 kinases were tested at the concentration of 100 nM across DiscoveRx KINOMEscan platform, and Fedratinib was used as the control (Fig. 4). Only 10 and 17 wild-type kinases are affected by compounds 1 and 3, respectively, while the number of wild-type kinases inhibited by Fedratinib is 34 (percent control <35%). Fedratinib shows binding to a wide range of kinases, whereas compounds 1 and 3 mainly target the TK group and have negligible effect on CMGC, CAMK, and AGC group. The results suggest that the macrocyclic compounds 1 and 3 have superior kinome selectivity profiles than Fedratinib.

Antiproliferative effects of compounds 1–3 on human erythroleukemia (HEL) and megakaryoblastic SET-2 cells, both of which are JAK2[V617F]-dependent, were also investigated. The results revealed that compounds 1 and 3 could suppress the proliferation of both cell lines, with compound 1 displaying comparable single-digit micromolar potency in comparsion to Fedratinib. Like other type I JAK2 inhibitors[47,48], compounds 1 and 3 increased the phosphorylation of

JAK2 at Y1007/8 site in HEL cells, but efficiently blocked the phosphorylation at Y221 site in a dose-dependent manner (Fig. 5), which is essential for JAK2 fully activation[49]. In addition, both the compounds significantly inhibited the activation of its downstream signaling molecule STAT3 and STAT5.

## In vivo pharmacokinetic analysis of compounds 1 and 3
The preliminary in vivo pharmacokinetic (PK) properties of compounds 1 and 3 and Fedratinib in mice following intravenous (iv, 5 mg/kg) and oral (po, 5 mg/kg) administration were investigated. The PK profiles are shown in Supplementary Fig. 8, and the analysis of PK parameters is summarized in Table 4. Compound 3 displayed overall superior PK properties than Fedratinib, except for the slightly lower bioavailability (F, 9.4% vs 11.7%). After oral dosing, compound 3 showed longer half-life ($T_{1/2}$, 10.07 vs 4.70 h) and higher systemic exposure ($AUC_{inf}$, 114.69 vs 50.19 h*ng/mL). Compared to Fedratinib, the macrocyclic compound 1 also displayed advantages in terms of oral PK properties, e.g., the higher systemic exposure (106.00 vs 50.19 h*ng/mL) and bioavailability (14.1% vs 11.7%). The holistically favorable PK profiles of the two macrocycles suggest that macrocyclization is an efficient strategy to improve the in vivo metabolic stability of Fedratinib.

## In vivo activities of compound 3
Overexpression of JAK2 has been reported in patients with inflammatory bowel disease (IBD), which means that inhibiting JAK2 may contribute to the treatment of IBD[50,51]. To assess the therapeutic potential of the macrocyclic JAK2 inhibitors for IBD, we established the dextran

## Table 3 | Structures and in vitro activities of compounds 1–3

| Compd | Structure | Enzyme Inhibitory Activity ($IC_{50}$, μM)[a] | Cellular Antiproliferative Activity ($IC_{50}$, μM)[b] | |
|---|---|---|---|---|
| | | | HEL | SET-2 |
| 1 | | 0.070 ± 0.006 | 2.41 ± 0.20 | 1.64 ± 0.03 |
| 2 | | 0.364 ± 0.007 | 7.34 ± 0.44 | >25 |
| 3 | | 0.006 ± 0.001 | 3.06 ± 0.39 | 11.09 ± 0.13 |
| Fedratinib | | 0.003 ± 0.001 | 1.34 ± 0.06 | 1.12 ± 0.08 |

[a]Data are mean ± SD, n = 3 parallel experiments. Source data are provided as a Source Data file.
[b]Data are mean ± SD, n = 2 independent experiments with three replicates each. Source data are provided as a Source Data file.

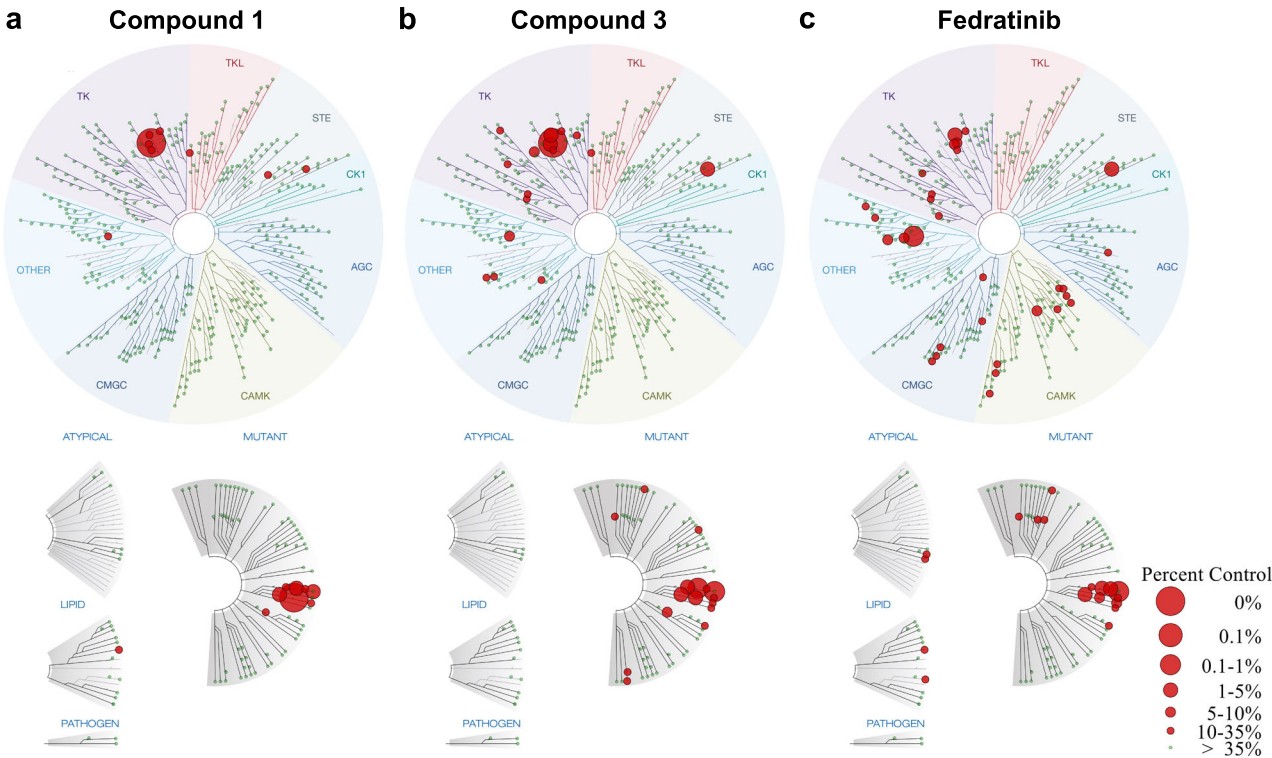

**Fig. 4 | Kinase selectivity profiles of compounds 1 (a) and 3 (b) and Fedratinib (c) against 468 kinases.** The affinity is defined as the percent of the DMSO control (percent control), where the lower value suggests stronger inhibition. Source data are provided as a Source Data file.

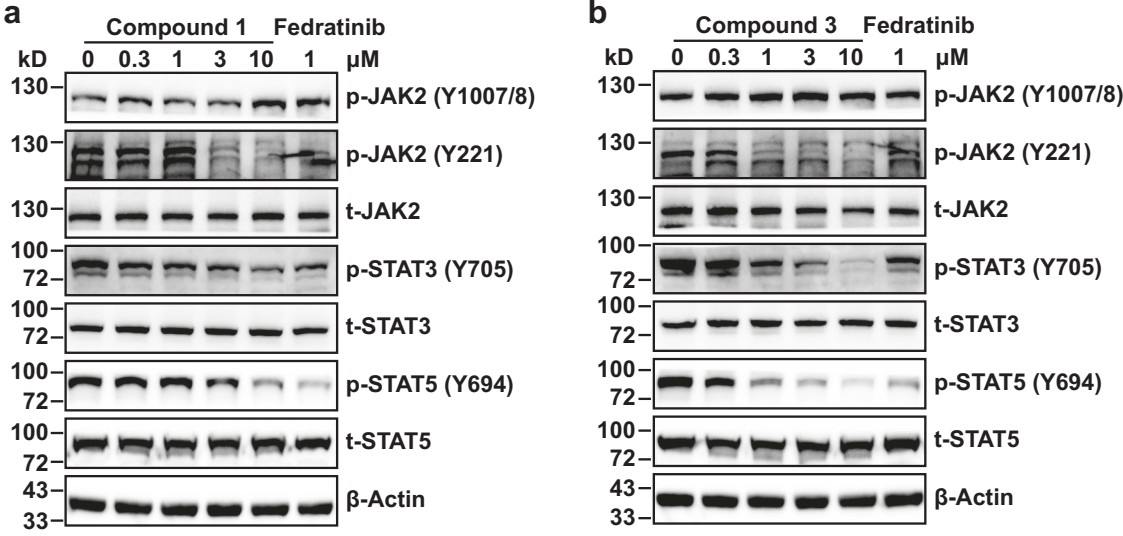

**Fig. 5 | Western blot analysis of compounds 1 (a) and 3 (b) for the effects on JAK2-STAT signaling pathways in HEL cells.** Fedratinib at 1 μM was used as the positive control. Three independent replicates were conducted obtaining similar results. Source data are provided as a Source Data file.

sulfate sodium (DSS)-induced colitis model[52]. The DSS colitis model recapitulates many clinical and pathological features of human IBD, such as bloody stools, weight loss, diarrhea, and inflammatory cells infiltration, and is widely used in IBD research as a preclinical model for initial studies. Salicylazosulfapyridine[53] was used as a positive control. After comprehensively considering the enzymatic and cellular activity, kinome selectivity, and PK properties, the macrocyclic compound **3** was selected for in vivo efficacy test. According to previous PK results, Fedratinib was administered at twice (10 mg/kg) the dose of

compound **3** (5 mg/kg). As shown in Fig. 6a, the administration of compound **3** and Fedratinib could alleviate the decrease of body weight caused by 3.5% (w/v) DSS. The disease activity index (DAI) scores of compound **3** and Fedratinib treating group were significantly decreased from day 8 (Fig. 6b). Moreover, it was found that compound **3** and Fedratinib treatment decreased the ratio of colon weight to length, a surrogate measure of colon inflammation, compared to the model group (Fig. 6c). The severity of colonic inflammation was then analyzed by H&E staining. Obviously, there was a significant

**Table 4 | PK parameters of compounds 1 and 3 and Fedratinib**

| Parameter[a] | Compound 1 | | Compound 3 | | Fedratinib | |
|---|---|---|---|---|---|---|
| Route | iv (5 mg/kg) | po (5 mg/kg) | iv (5 mg/kg) | po (5 mg/kg) | iv (5 mg/kg) | po (5 mg/kg) |
| $T_{1/2}$ (h) | 5.35 | 4.47 | 0.52 | 10.07 | 0.3 | 4.70 |
| $T_{max}$ (h) | 4.00 | 6.00 | 0.083 | 4.00 | 0.083 | 1.00 |
| $C_{max}$ (ng/mL) | 76.57 | 10.35 | 351.23 | 6.64 | 275.23 | 6.81 |
| $AUC_{inf}$ (h*ng/mL) | 752.47 | 106.00 | 443.20 | 114.69 | 226.74 | 50.19 |
| $T_{last}$ (h) | 7.63 | 7.99 | 4.63 | 15.94 | 2.09 | 7.45 |
| F (%) | | 14.1 | | 9.4 | | 11.7 |

[a]$T_{1/2}$ elimination half-life, $T_{max}$ time of the maximum observed plasma concentration, $C_{max}$ maximum observed plasma concentration, $AUC_{inf}$ area under the plasma concentration–time curve from time zero to infinity, $T_{last}$ time of last measurable concentration, F bioavailability.

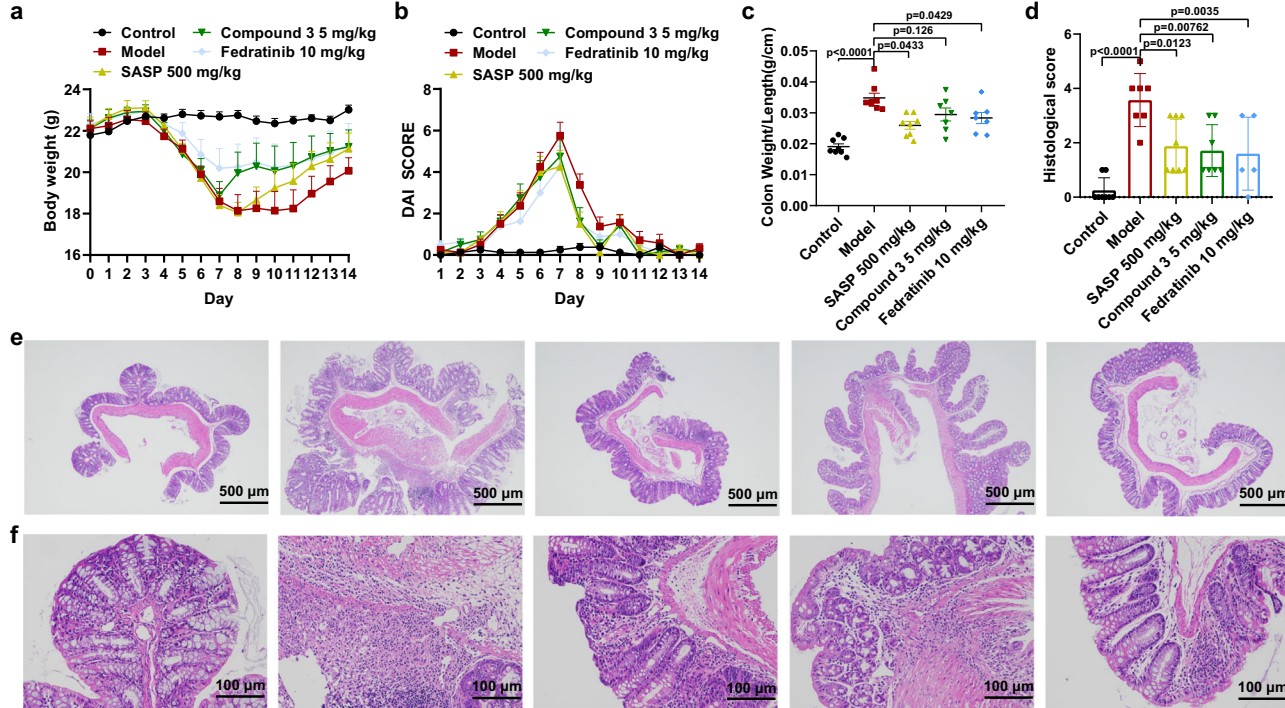

**Fig. 6 | Effects of compound 3 on DSS-induced colitis mice. a** The daily weight changes in each group (n = 8 mice per group). **b** The DAI score change curves during experiment (n = 8 mice per group). **c** The ratio of colon weight to length in each group. Control, n = 8 colon tissues; model, n = 8 colon tissues; SASP, n = 8 colon tissues; compound 3, n = 7 colon tissues; Fedratinib, n = 7 colon tissues. **d** The histological scores based on Ameho criteria[62] in each group. Control, n = 8 colon tissues; model, n = 7 colon tissues; SASP, n = 8 colon tissues; compound 3, n = 7 colon tissues; Fedratinib, n = 5 colon tissues. **e** Representative H&E-stained tissue sections illustrating the features of colon from each group. Scale bar, 500 μm. **f** Representative H&E-stained tissue sections illustrating the features of colon from each group. Scale bar, 100 μm. Values are shown as means ± SEM, and statistical analysis was performed with one-way ANOVA followed by LSD post-hoc test. Source data are provided as a Source Data file.

inflammatory response in the model group, which was characterized by significant inflammatory cells infiltration, goblet cell loss, near-complete crypt loss, reactive epithelial hyperplasia, sub-mucosal edema, and irregular colonic villi. In contrast, compound **3** and Fedratinib treated mice exhibited less inflammatory cell infiltration in colon tissue, intact colonic architecture with less apparent ulceration, and lower histological scores (Fig. 6d–f). Collectively, these results indicate that compound **3** and Fedratinib are able to ameliorate the symptoms in DSS-induced murine colitis, and that compound **3** showed comparable therapeutic efficacy to Fedratinib at a lower dose.

As a JAK2 inhibitor, the macrocyclic compound **3** has shown preliminary therapeutic efficacy for IBD in our initial study using the acute DSS colitis model. However, there are still many issues to be addressed by extensive research. From the mechanistic point of view, JAK2 deeply engages in the signaling of hematopoietic cytokines and hormones, and the potential side effect of myelosuppression renders

JAK2 controversial as a therapeutic target for IBD[54]. Therefore, the indicators associated with myelosuppression should be carefully monitored for in vivo test, in order to clarify the efficacy and safety of selective JAK2 inhibitors for IBD. Meanwhile, chronic IBD animal models are recommended to better mimic the chronic pathological conditions in humans.

## Discussion
To tackle the automatic macrocyclization problem by leveraging the benefits of deep learning, we developed the Macformer model on the basis of Transformer architecture. Starting from an acyclic structure represented as SMILES strings and labeled with two attachment points, Macformer is dedicated to automatically generate corresponding cyclized analogs. With data augmentation skills using substructure-aligned randomized SMILES notations, Macformer is able to capture the hidden connections between the source linear and target

macrocyclic SMILES sequences of the ChEMBL training dataset and efficiently produce macrocycles with chemical diversity and novelty on both internal ChEMBL and external ZINC test dataset. The excellent performance and generalization capability of Macformer imply its potential for the design of macrocyclic compounds, thus expanding the application of deep learning technology in the field of drug discovery.

The ultimate goal of computational method development is to assist in practical drug design process. Following this philosophy, Macformer was utilized to design macrocyclic JAK2 inhibitors, with the core structure of Fedratinib as the original acyclic scaffold. The macrocycles generated by Macformer were docked into the ATP binding site of JAK2 to further evaluate their interactions with the target, which were used as an import criterion for subsequent compound selection. Among the 218 generated novel macrocycles with distinct linkers, three were synthesized and tested for their biological activities. Among them, compounds **1** and **3** exhibited inhibition activities against JAK2 at both enzymatic and cellular level, and displayed an improved selectivity profile against 468 kinases and favorable PK properties than Fedratinib. Additionally, compound **3** manifested in vivo anti-inflammatory effect on DSS induced murine colitis at a lower dose than Fedratinib. The prospective case study validates the practicability of Macformer, which can provide potential macrocyclic scaffolds for further development of drug candidates targeting the JAK2 kinase as well as other drug targets. It is expected that, as a powerful complement to the traditional macrocyclization method, Macformer will play a valuable role in the design of macrocyclic drug candidates.

## Methods

### Data preparation

The macrocyclic dataset was collected from ChEMBL database[30], and only compounds satisfying the following criteria were retained: 1) contain only one macro ring with 12 or more atoms; 2) molecule type is labeled as "small molecule"; 3) bioactivity data is not empty; 4) canonical SMILES strings length is inferior to 200. The canonical SMILES representation was generated using RDKit, and the stereochemical information was removed for simplification. After filtration of duplicate structures, 18,357 unique macrocycles were derived.

To acquire matched acyclic-macrocyclic pairs, the macrocycles were fragmented to two substructures through simultaneously cutting two single bonds of the macro ring. The substructure with more heavy atoms was classified as the acyclic analog and the other the linker. For each macrocyclic compound in the dataset, the fragmentation process would create multiple combinations of acyclic analog and corresponding linker. From the perspective of synthetic accessibility, the structurally simple macrocycle linker should be preferrable. Consequently, the linkers were filtered according to the following criteria: 1) contain only one ring structure with 6 atoms or less; 2) the number of heavy atoms on the shortest path is restricted within the range of 3–9; 3) the ratio between the number of heavy atoms on the shortest path and the whole linker is more than 0.6; 4) the ratio between the number of heavy atoms of the linker and the original macrocyclic compound is less than 0.25. The SMILES strings of the acyclic analog were also canonicalized using RDKit, with the two cutting points marked with dummy atoms. The information in terms of linker length was added as a token prepended to the sequence. By corresponding linker length tokens[55], the resulting dataset containing a total of 237,728 unique matched acyclic-macrocyclic pairs was randomly split into a training set (80%), a validation and a test set (10% for each).

We collected 486 bioactive macrocycles from ZINC database, all of which are not present in ChEMBL dataset. Then 5551 acyclic-macrocyclic SMILES pairs were extracted from them as external test dataset using the same data processing protocol.

### Data augmentation

Multiple randomized SMILES were generated using RDKit as a means to achieve data augmentation. A molecule can be represented as a 2D graph, from which linear SMILES notations can be derived through enumerating nodes of the graph following a certain topological ordering. By setting the doRandom parameter of the MolToSmiles function as True, RDKit would randomly select a starting node and the topological path to enumerate the molecule graph, then randomized SMILES of the input acyclic scaffolds were generated. The randomized SMILES of the target macrocycles were generated in a restricted manner. After substructure matching by RDKit, the indices of the macrocycle's atoms that match the acyclic substructure query would be returned. These indices were placed first, followed by the atomic numbers of other structural moiety, in order to reorder the atoms of the macrocycle. The randomized SMILES of the target macrocycle was finally obtained according the new atomic numbers.

### Macformer

The model was implemented based on Transformer architecture[31], which has a stepwise autoregressive encoder-decoder architecture. Both source and target SMILES sequences are tokenized and embedded into trainable matrix, with the embedding vector size set as 256 for each token. Besides, the sine and cosine functions are utilized as positional encoding to indicate the position of different tokens in the sequence:

$$PE_{(pos,2i)} = \sin\left(\frac{pos}{10000^{\frac{2i}{d_{emb}}}}\right), PE_{(pos,2i+1)} = \cos\left(\frac{pos}{10000^{\frac{2i}{d_{emb}}}}\right) \quad (1)$$

where *pos* is the position and *i* denotes the iterator used to construct this vector, which runs from 0 to $d_{emb}/2$. The positional encodings are added to the token embeddings, and each sequence is finally represented as follows:

$$\mathbf{X} = (x_1, x_2, \ldots, x_n) \quad (2)$$

where $x_i$ is the vector of the *i*th token (with positional vectors added) in a sequence containing *n* tokens.

The embedding matrices of the source sequences are fed into the encoder to generate a latent representation $\mathbf{L} = (l_1, l_2, \ldots, l_n)$ to initialize the decoding process. Both encoder and decoder are stacked by identical layers. Each encoder layer consists of a multi-head attention sublayer and a positional feed forward network sublayer. Unlike the encoder, an extra encoder-decoder attention sublayer is inserted to each decoder layer, which performs multi-head attention over the output of the encoder stack and helps the decoder focus on appropriate places in the input sequence.

The multi-head attention mechanism allows the encoder and decoder to peek at different tokens simultaneously, thus the transformer mode can successfully cope with long-range dependencies. A multi-head attention unit comprises eight parallelly running scaled-dot attention layers in this study, which are concatenated and projected into the final values. The scaled-dot attention layer takes three matrices as input: the matrix **Q** packed with a set of queries, the matrix **K** with keys, and the matrix **V** with values. The attention is computed as follows:

$$attention(\mathbf{Q}, \mathbf{K}, \mathbf{V}) = softmax\left(\frac{\mathbf{Q}\mathbf{K}^{\mathrm{T}}}{\sqrt{d_k}}\right)\mathbf{V} \quad (3)$$

where $d_k$ is a scaling factor depending on the size of the weight matrices.

At the end of the Transformer model, linear transformation and softmax function are successively applied to convert the decoder

output to predicted next-token probabilities. For a particular source sequence, the training objective is to minimize the gap between the predicted sequence and its corresponding target sequence, which is estimated by cross-entropy loss function:

$$Loss = -\sum_{i=1}^{k} y_i \log(m_i) \qquad (4)$$

where $k$ is the token number of the target sequence, and $y_i$ and $m_i$ are the ground truth and predicted values at the $i$th position of the target sequence, respectively.

In our study, the Transformer model was constructed with four encoder and decoder layers of size 256, respectively, resulting a total of 12 M trainable parameters. A dropout rate of 0.1 was used in both dense and attentional layers to preform regularization. The model was optimized using the Adam optimizer[56] with $\beta_1 = 0.9$ and $\beta_2 = 0.998$, and the learning rate varied using 8000 warm up steps during the course of training. The batch size was set to 2048 tokens and the gradients were accumulated over four batches before updating parameters. The model was trained 200,000 steps on one GPU (NVIDIA TESLA V100). One checkpoint was saved every 10,000 steps and then was applied for model validation on the validation set. Teacher forcing strategy was adopted during training and validation courses, hence the output token was predicted based on the ground truth value from previous timestep[57]. All experiments were carried out with the PyTorch version of OpenNMT[58].

Beam search algorithm[33] was adopted to decode the source sequences of the test datasets. As the predicted sequences are constructed, the beam search expands all possible next tokens while keeping track of the top-$k$ sequences based on the product of the probabilities of each token.

## MacLS
To ensure that macrocycles are constructed using the same data as that of Macformer, 9243 unique linkers from the ChEMBL training dataset were used to construct the 3D linker database. A maximum of 20 low-energy conformations were generated for each linker and a total of 163,924 structures were acquired. This process was implemented using RDKit, following the procedure proposed by ref. 59. The conformations of the acyclic and macrocyclic compounds were generated using the same method, and top 3 low-energy conformations were reserved for each chemical structure. To speed up the macrocyclization process, linkers were first filtered through geometric criteria. For the 3D acyclic structures and linkers, two distance parameters were calculated, one was between the two leaving atoms on the two attachment vectors and the other between their adjacent atoms. The dihedral angle of the two attachment vectors was also calculated as an additional parameter (Supplementary Fig. 2). The distance threshold between the acyclic structure and the linker is set to 0.5 Å, and the dihedral angle threshold was set to 20°. After superimposing the attachment vectors of the given acyclic structure and that of the linkers satisfying the geometric constraints, the RMSD values between the atoms of the attachment vectors were calculated (https://github.com/charnley/rmsd), and top 10 linkers were used to construct macrocycles.

## Model evaluation metrics
The performances of Macformer and MacLS were evaluated by the metrics widely used in previous molecular generation work[60].

**Recovery** is the percentage of correctly predicted target macrocycles of the test dataset.

**Validity** is the percentage of generated chemically valid molecules.

**Uniqueness** is the percentage of unique molecules in the generated valid molecules.

**Novelty_mol** is the percentage of the novel molecules, which are not present in the training set, in the generated validly unique molecules.

**Novelty_linker** is the percentage of the novel molecules, which have novel linkers that are not present in the training set, in the generated validly unique molecules.

**Macrocyclization** is the percentage of macrocycles in the generated validly unique molecules, and it is a distinct metric for the macrocyclization method.

## Molecular docking
The crystal structure of JAK2 binding with Fedratinib (PDB code 6VNE) was derived from the Protein Data Bank and prepared using the Protein Preparation Wizard of Maestro v10.1. The grid-enclosing box was placed on the centroid of the crystallographic ligand and a scaling factor of 0.8 was set to van der Waals radii with partial atomic charges of less than 0.15 to soften the nonpolar parts of the receptor. The three-dimensional structures of compounds were generated and minimized with Ligprep v3.3 module. Standard precision (SP) approach of Glide was adopted to dock the molecules into the binding site with the default parameters, and only the top one pose was retained for each molecule.

## Enzyme assay
JAK kinase activity assays were performed using Z'-LYTE™ kinase assay kit (Life Technologies, pv4122). Enzyme reactions include 10 μL volumes of 1 × kinase buffer (50 mM HEPES pH 7.5, 10 mM MgCl$_2$, 1 mM EGTA, 0.01% Brij-35) along with 25 μM ATP, 0.05–0.42 ng JAK2, 2 μM peptide substrate (Tyr06), and various concentrations of compounds. The mixture was added into the 384-well microplate, shaken gently, and then incubated at room temperature for 1 h. Subsequently, 5 μL development solution was added to the well for incubating another 1 h. Finally, 5 μL stop reagent was added to stop the reaction. The fluorescence was measured with excitation at 400 nm, and emission at 445 and 520 nm. IC$_{50}$ values were calculated using GraphPad Prism v8.0.3, and three parallel experiments were performed.

## Cell proliferation assay
The human erythroid leukemia (HEL 92.1.7) cells were purchased from the American Type Culture Collection (ATCC, TIB-180), and the human megakaryoblastic (SET2) cells were obtained from German Collection of Microorganisms and Cell Cultures (DSMZ, ACC 608). The anti-proliferative activities of compounds were evaluated using WST-8 cell counting kit-8 (Elabscience). HEL and SET-2 cells were seeded at 5000 cells/well in 70 μL RPMI-1640 medium (Hyclon) with 10% FBS (Gibco) to the 96-well plate and incubated overnight at 37 °C with 5% CO$_2$. The compounds were serially diluted in RPMI-1640 medium, and cells were treated separately with 30 μL various concentrations of compounds for 72 h. The final DMSO concentration in the culture wells was 0.1%, which had no effect on the cell viability. Finally, 10 μL of CCK-8 solution was added into the wells. 4 h later, the absorbance values at 450 nm were recorded. IC$_{50}$ values were determined using GraphPad Prism v8.3.0, and three parallel experiments were performed.

## Western blot analysis
HEL cells were added into six-well plates ($2 \times 10^6$ cells/well) and then put into the incubator (37 °C, 5% CO$_2$). After 24 h, the tested compounds were diluted in RPMI-1640 medium (final DMSO concentration = 0.1%) with different concentrations (0 μM, 0.3 μM, 1 μM, 3 μM, and 10 μM) was added separately into the wells, and incubated for another 0.5 h, then cells were collected and lysed. Proteins from each sample were isolated by SDS-PAGE and transferred to PVDF membrane. The membranes were blocked in 5% milk (TBST) for 1 h at room temperature, and subsequently incubated with the indicated primary antibody in blocking buffer overnight at 4 °C. Primary antibodies were

used as follows: p-Y1007/8-JAK2 (#3771, CST, 1:1000), p-Y221-JAK2 (#11150, SAB, 1:500), p-Y705-STAT3 (#11045, SAB, 1:1000), p-Y694-STAT5 (#13386, SAB, 1:2000), STAT3 (10253-2-AP, Proteintech, 1:4000), STAT5 (12071-1-AP, Proteintech, 1:4000), JAK2 (E-AB-70193, Elabscience, 1:2000) and β-Actin (E-AB-20034, Elabscience, 1:10000). Then the membranes were washed with TBST (3 × 5 min), incubated with HRP-conjugated secondary antibody for 1 h at room temperature, washed again, and then exposed by chemiluminescence method using the enhanced ECL immunoblotting system (Tanon, Shanghai, China). All the experiments were repeated in triplicate. Blot bands were quantified by densitometry using Image J software v1.51.

### Kinase selectivity profile
The kinase selectivity profile was performed by using the DiscoveRx KINOMEscan platform. The compounds were screened at a concentration of 100 nM against a panel of 468 kinases. Test compounds were prepared as 40× stocks in 100% DMSO and directly diluted into the assay. The results were defined as a percentage of signal between the negative (DMSO, 100% control) and the positive (control compound, 0% control) control, which was calculated as follows: percent control = [(test compound signal − positive control signal)/(negative control signal − positive control signal)] × 100.

### In vivo PK study
The PK parameters of the compounds in male BALB/c mice (6–7 weeks, 20–25 g, Shanghai SLRC laboratory animal Co. Ltd) were conducted by Hangzhou Leading Pharmatech Co., Ltd. The mice were kept in a temperature-control room (22–25 °C, relative humidity 52–63%) with 12 h dark/light cycles, and were allowed free access to food and water for 3 days to be adapted to the environment before experiment. The 1 mg/mL dosing solutions of compound **3** and Fedratinib were prepared in the solubilizing vehicle (5% DMSO/30% PEG400/65% saline) for intravenous and oral administration. The same vehicle was used for compound **1** in oral administration, while 5% DMSO/15%(100%Solution)/80% saline was used in intravenous administration. The mice were separately administered to a group of three mice per time point for intravenous (5 mg/kg) or oral administration (5 mg/kg). Blood samples were collected at 0.083, 0.25, 0.5, 1, 2, 4, 8, and 24 h after intravenous administration, and at 0.25, 1, 2, 3, 4, 6, 8, and 24 h after oral administration. Then the samples were separated by centrifugation and analyzed by LC-MS/MS (XEVO TQ-S) to determine the plasma drug concentrations. The PK parameters were calculated using the noncompartment model with Phoenix WinNonLin v8.0. The animal experimental procedures were approved by the Medicine Research Ethics Committee of Hangzhou Leading Pharmatech Co., Ltd.

### In vivo efficacy study
Male BALB/c mice (6 weeks, ~20 g) were obtained from Shanghai Sippr-BK laboratory animal Co. Ltd. The mice were kept in a temperature-control room (25 °C, relative humidity 40–60%) with 12 h dark/light cycles, and were allowed free access to food and water for 1 week to be adapted to the environment before experiment. All animal experimental procedures were approved by the Medicine Research Ethics Committee of East China University of Science and Technology.

Acute colitis was induced by replacing the drinking water with 3.5% DSS (MW: 36,000–50,000, YEASEN) for 7 days (from day 1 to day 7), during which healthy control mice received normal drinking water. Model mice were divided into four groups and eight mice were included in each group on day 8. The compounds to be evaluated were formulated in the solubilizing vehicle (5% DMSO/30% PEG400/65% saline) and given intragastrically, a routine administration method that was also used in the previously repored in vivo test on mice of Fedratinib[61], to different groups at a fixed time from day 8 to day 14. Meanwhile, solvent was given intragastrically to the control group and

model group. On day 15, the mice were sacrificed, and the length and weight of each colon were measured.

Body weight, the stool consistency, and the presence of blood in feces were recorded daily in the morning. The DAI was assessed by the criteria: 0, body weight loss less than 1%, normal stool, no rectal bleeding; 1, body weight loss 1–4.99%, softer stool, weak rectal bleeding; 2, body weight loss 5–10%, moderate diarrhea, visual blood in stool; 3, body weight more than 10%, diarrhea, fresh rectal bleeding. The maximum score was 9, which was the sum of the scores. About 1-cm colon tissue, which is 0.5 cm away from the anal margin, was collected, and the ileocecal area was used for histopathological examination. After H&E staining, the tissues were scored under single-blind conditions to evaluate the levels of inflammation and tissue damage in the colons following the criteria proposed by ref. 62. Statistical analyses were performed using SPSS v24.0.

### Reporting summary
Further information on research design is available in the Nature Portfolio Reporting Summary linked to this article.

## Data availability
The crystallographic structure of Fedratinib in complex with JAK2 used in this study is available in the PDB database under accession code 6VNE. The acyclic-macrocyclic SMILES pairs extracted from ChEMBL and ZINC database, respectively, and the pretrained models generated in this study are available at GitHub https://github.com/yydiao1025/Macformer. Source data are provided with this paper.

## Code availability
The source code of Macformer and associated data preparation python v3.6.10 scripts are available at GitHub (https://github.com/yydiao1025/Macformer)[63].

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

## Acknowledgements

This work was supported in part by the National Key Research and Development Program of China (2022YFC3400501); and the National Natural Science Foundation of China (81825020 and 82150208); H.L. was also sponsored by the National Program for Special Supports of Eminent Professionals and the National Program for Support of Top-notch Young Professionals.

## Author contributions

Y.D. developed the Macformer method and wrote the manuscript; D.L., R.Z., H.B., W.L., and Z.Z synthesized the macrocyclic compounds; X.Z. analyzed the stability of the synthesized macrocyclic compounds in solution; H.G., Z.C., and L.Z. evaluated the activities of the compounds against JAK2 at enzymatic and cellular level; K.J., R.B., and R.W. carried out the in vivo experiments; K.Z. and Q.H. helped to revise the manuscript; H.L. designed the whole project and revised the manuscript.

## Competing interests

The authors declare no competing interests.
