## [Peer Review File · Nature Communications]

REVIEWER COMMENTS

Reviewer #1 (Remarks to the Author):

The enzymatic assays appear to be carried out appropriately and highlight similar levels of activity to Fedratinib, particularly for compound 3. The selectivity was assessed for compound 3 and indicates a high degree of selectivity. How does this compare to Fedratinib? The proliferation effects are also similar to Fedratinib. The rationale for the use of the colitis model is not explained and some additional information is required. It would be best to include a graph with the histology score and to improve the readability of the scale on the histology images. Is there a comparison of the activity with Fedratinib? The rationale for the method of administration of the compound should also be explained. What was the sample dissolved in for the in vitro and in vivo experiments? Was the stability of the different compounds assessed and compared to Fedratinib? This is generally a critical point in macrocyclic studies.

Reviewer #2 (Remarks to the Author):

Yanyan Diao et al. are reporting a generative modeling study of developing macrocyclic compounds in this manuscript. Indeed, macrocyclic molecules are playing an important role in the modern drug discovery. Using deep learning to realize the automated generation of macrocyclic compounds is a beneficial addition to the application scope of generative chemistry. Overall the manuscript is well-organized with a detailed description on the method development and prospectively designed experimental validations. However, the reviewer does have some major and minor concerns to point out.

1. The prospective case study on developing JAK2 inhibitor doesn't support the claimed advantages of adapting macrocyclic compounds. Authors claimed that "Comparing with the linear analogues, macrocycles tend to possess pre-organized restricted conformations and extended contacts with targets, thus potentially achieving improved binding affinities, better selectivities and superior pharmacological characteristics." In the enzyme inhibitory assay and the cellular antiproliferative assay as shown in table 3, newly designed cyclic compounds 1-3 do not exhibit the improvement to the control of Fedratinib. While in the following of kinase selectivity test and the in vivo activity test, the control of Fedratinib is simply missing. Thus, from the presented data, there is no evidence of "improved binding affinities, better selectivities, and superior pharmacological characteristics".

2. Then the value of the presented deep learning macrocyclic modification in this manuscript shrinks to a method that can bypass intellectual property restrictions of known active molecules. In this scenario, the academic significance is compromised but the significance to the industrial applications still remains. Here comes the second major concern. When developing JAK2 inhibitors, what is the rationale behind transferring Fedratinib into a macrocyclic compound? The ligand structural and protein structural foundations are missing. Or is bypassing the intellectual restrictions of Fedratinib an objective?

The following are more specific points.

3. In the Introduction, authors claimed “Due to the distinctive features, macrocycles are regarded as a privileged chemotype for targeting challenging proteins that are hardly tractable by traditional small molecule drugs⁷, bridging the gap between small molecules and large biologics”. Generally, the reviewer agrees on that. But not all challenging proteins can be targeted by macrocycles. A brief but explicit discussion can be beneficial.

4. In the section of Model overview. “the number of macro ring is inferior to 1”. What is the definition here of a macro ring? Is it by counting the number of heavy atoms?

5. Models were trained with different augmentation levels. The method and the strategy of selecting starting atoms and the direction of writing that structure into the SMILES string should be specified. The augmentation is not enumerating all possibilities but is defined with 2X, 5X, and 10X.

6. As authors pointed out that data augmentation helps in learning the basic syntax of the chemical language to produce chemically meaningful SMILES strings. Could a pre-trained model be produced first by learning general ChEMBL / ZINC molecules? Then the transfer learning process can be performed to add specificity to the macrocyclic molecules.

7. Zinc macrocycles are having lower M.W. than ChEMBL macrocycles. But it is inappropriate of claiming them to be “more drug-like”. M.W. between 500~800 can hardly be considered as “drug-like” molecules. The length of SMILES strings is usually correlates to the M.W. But we don’t consider the length of SMILES strings as a “drug-like property”.

8. In Fig.2, t-SNE plots was reported for generated and raw ChEMBL compounds. Adding generated ZINC macrocyclic compounds can be interesting to compare as those compounds were never seen during the training.

9. t-SNE plots also suggested that the generation can never jump out of the known chemical space. In the Introduction authors commented that Wagner et al. and Sindhikara et al.'s methods are limited to the collection of fragment linkers. But it is the same story with authors' method here, the creativity of the generation is limited to the original training set.

Overall, the reviewer would suggest a major revision if above-mentioned comments and questions can be appropriately addressed.

Reviewer #3 (Remarks to the Author):

This manuscript describes the development and application of a computational method termed "Macformer" to propose chemical linkers suited to convert acyclic ligands into macrocyclic ones. The authors indicate that the software tool takes advantage of deep learning principles. They further indicate that they have trained the software and applied it then to propose cyclization linkers of the approved drug bedratinib being an acyclic small molecule JAK2 inhibitor. The software tool proposed 281 macrocyclic compounds of which they synthesized three and tested them in vitro. One was highly potent, showed high target selectivity and proved active in vivo.

Reading the title and abstract, I was excited to read the study, but I got disappointed by the work the more I read it in detail. In particular, I was shocked when I found data in literature in which the linker that the authors "found" by the new software tool was already reported, and is present in an approved macrocyclic JAK inhibitor drug (pacritinib). I describe my four major concerns in the following, all being alone a reason to decline the work. I do not recommend publication.

1. Difficulty to understand computational methods and concepts behind:

I found it hard to understand the computational methods described. The concepts are not well described and are rather cryptic. There are many examples of studies that describe complex computational methods but manage to describe the concepts in an understandable way (e.g. the recent Cell paper by the Baker lab which I read and enjoyed as it clearly describes the concept(s) of the new approach). Without making the concepts/principles understandable for the medchem community, this work is of little value, and for reviewers like me, it is not possible to evaluate the concepts behind the software tool.

2. Taking advantage of structural information:

It appears to me that the software tool does not take advantage of the structural information to propose linker structures suited to cyclize acyclic ligands. In particular for the applied example, there is an X-ray structure of JAK2 with Fedratinib bound, which can be taken to measure the distance between the two points that need to be connected, as well as their orientation. A computational tool that does not use this information will never be superior to a conventional approach where structure information is taken into account and potential linkers are modeled.

3. Comparison with conventional methods:

The authors do not side-by-side compare their tool with conventional methods. Reading the literature, I found that several groups and companies have already macrocyclized acyclic ligands of JAK kinases, and a comparison would be required. For the reasons described in point 2, it is essentially impossible that the developed tool can be better than conventional methods that take advantage of available structure information.

4. The "identified" linker in compound 3 is already reported in literature:

The linker of the best one of the three compounds tested is already reported in literature. It is present in the approved drug Pacritinib, which is a macrocyclic JAK inhibitor (!). The authors do not mention this inhibitor in their report and neither describe relevant efforts to macrocyclize acyclic JAK inhibitors by other methods. An audience seeing the results of this work would immediately think that the authors have been cheating and simply have stolen the linker from the approved drug. They would not believe that it was truly identified by the new software tool. Personally, I think that presenting the data as done herein, without mentioning that this linker was found in other macrocyclic ligands of JAK targets (and even in an approved drug), is not honest and poor scientific practice.

Reviewer #4 (Remarks to the Author):

The mechanisms used for drug design and selection look interesting. The colitis part in this manuscript is weak, however. First, I think that Jak2 is certainly not considered as a good target for gut inflammation in humans given the issues on myelotoxicity and thus studies in humans currently focus on Jak1 and to a lesser extent on Tyk2. Second the mouse model used is a model of acute intestinal injury and does not mimic chronic inflammation in humans. Finally, the authors showed prevention studies but did not demonstrate the therapeutic value of their compound.

To Reviewer #1 (Remarks to the Author):

The enzymatic assays appear to be carried out appropriately and highlight similar levels of activity to Fedratinib, particularly for compound 3. The selectivity was assessed for compound 3 and indicates a high degree of selectivity. How does this compare to Fedratinib? The proliferation effects are also similar to Fedratinib. The rationale for the use of the colitis model is not explained and some additional information is required. It would be best to include a graph with the histology score and to improve the readability of the scale on the histology images. Is there a comparison of the activity with Fedratinib? The rationale for the method of administration of the compound should also be explained. What was the sample dissolved in for the in vitro and in vivo experiments? Was the stability of the different compounds assessed and compared to Fedratinib? This is generally a critical point in macrocyclic studies.

Response: We appreciate the reviewer's helpful comments on our work. Accordingly, we have revised our manuscript fully taking into account the reviewer's concerns.

1. The selectivity was assessed for compound 3 and indicates a high degree of selectivity. How does this compare to Fedratinib?

Response: Thanks for the reviewer's constructive advice. We have added the kinase selectivity experiment in the revised manuscript. The result demonstrated the improved kinase selectivity profile of compound 3 than Fedratinib. Only 17 wild-type kinases are affected by compound 3, while the number of wild-type kinases inhibited by Fedratinib is double (34). Fedratinib shows binding to a wide range of kinases, whereas compound 3 mainly targets the TK group and has negligible effect on CMGC, CAMK, and AGC group.

2. The proliferation effects are also similar to Fedratinib. The rationale for the use of the colitis model is not explained and some additional information is required.

Response: The rationale for the use of the colitis model has been explained in the revised manuscript. Overexpression of JAK2 has been reported in patients with inflammatory bowel disease (IBD), which means that JAK2 may become a therapeutic target for IBD. To assess the therapeutic effects of compound **3** and Fedratinib, we established the dextran sulfate sodium (DSS)-induced colitis model. The DSS colitis model recapitulates many clinical and pathological features of human IBD, such as bloody stools, weight loss, diarrhea, and inflammatory cells infiltration, and is widely used in IBD research as a preclinical model.

3. It would be best to include a graph with the histology score and to improve the readability of the scale on the histology images.

Response: According to the reviewer's kind suggestion, the histological scores based on Ameho criteria has been added (Fig. 7d) in the revised manuscript.

4. Is there a comparison of the activity with Fedratinib?

Response: We redid the *in vivo* efficacy evaluation test and added a fedratinib control group. The added pharmacokinetic study displayed that Fedratinib has shorter half-life ($T_{1/2}$, 4.70 vs 10.07 h) and lower systemic exposure (AUC_{inf} , 50.19 vs 114.69 h*ng/mL) after oral dosing. So Fedratinib was administered at twice (10 mg/kg) the dose of compound **3** (5 mg/kg). The *in vivo* test showed that both compound **3** and Fedratinib are able to ameliorate the symptoms in DSS-induced murine colitis, but compound **3** (5 mg/kg) showed comparable therapeutic efficacy to Fedratinib (10 mg/kg) at a lower dose.

5. The rationale for the method of administration of the compound should also be explained.

Response: Compound **3** and Fedratinib was given to the mice by oral gavage in our study. The oral gavage is a routine administration method that was also used in the previously reported *in vivo* test on mice of Fedratinib¹. Because of the low solubility, the two

compounds were formulated in the solubilizing vehicle (5% DMSO/30% PEG400/65% saline). The solvent was also given intragastrically as a control group, which showed safety to the mice in the test.

6. What was the sample dissolved in for the in vitro and in vivo experiments?

Response: The solvent information for the in vitro and in vivo experiments has been added in the revised manuscript. Stock solutions of test compounds were made in dimethylsulfoxide (DMSO), and subsequently diluted in different solvent for different experiments. The compounds were diluted in kinase buffer (50 mM HEPES pH 7.5, 10 mM MgCl₂, 1 mM EGTA, 0.01% Brij-35) of Z'-LYTETM kinase assay kit for enzyme assay. In cell proliferation, Western Blot, and kinome selectivity assay, the compounds were diluted in RPMI-1640 medium. In in vivo pharmacokinetic and efficacy evaluation test, test compounds were prepared in the solubilizing vehicle (5% DMSO/30% PEG400/65% saline).

7. Was the stability of the different compounds assessed and compared to Fedratinib? This is generally a critical point in macrocyclic studies.

Response: We first evaluated the stability of compound **2**, compound **3**, and Fedratinib in DMSO stock solution. Test compounds were incubated at room temperature and 60 °C, respectively, for 72 h, and their purities (%) were determined using HPLC equipped with a CORTECS C18 column (4.6*50 mm, 2.7 µm particle size) and a UV/VIS detector setting of $\lambda=210$ nm. All compounds were eluted with the two solvent systems (ammonium formate as organic phase in Method I and CH₃OH as organic phase in Method II) at a flow rate of 0.3 ml/min. Each experiment repeated for three times and the results are represented as mean±SD. As illustrated in Table R1, all the three compounds showed comparable high stability in DMSO.

Table R1. Comparison of chemical stability of different compounds at room temperature and 60 °C, respectively.

Time (h)	Room Temperature			60 °C		
	Comp. 2	Comp. 3	Fedratinib	Comp. 2	Comp. 3	Fedratinib
0	99.65±0.05	99.68±0.05	99.36±0.06	99.59±0.13	99.78±0.07	98.91±0.10
2	99.65±0.05	99.77±0.01	99.11±0.14	99.60±0.06	99.46±0.04	98.91±0.10
24	100.00±0.00	99.68±0.07	99.99±0.00	100.00±0.00	99.64±0.04	98.89±0.17
48	99.99±0.01	99.63±0.02	99.08±0.02	99.61±0.01	99.66±0.06	100.00±0.00
72	99.50±0.06	99.64±0.10	99.57±0.06	99.43±0.09	99.78±0.03	98.96±0.15

Meanwhile, the *in vivo* pharmacokinetic properties of compound **3** and Fedratinib in mice following intravenous (iv, 5 mg/kg) and oral (po, 5 mg/kg) administration were investigated. As shown in Table 4 of the revised manuscript, compound **3** displayed overall superior PK properties than Fedratinib. After oral dosing, compound **3** showed longer half-life ($T_{1/2}$, 10.07 vs 4.70 h) and higher systemic exposure (AUC_{inf} , 114.69 vs 50.19 h*ng/mL). The holistically favorable PK profile of compound **3** suggests that macrocyclization of Fedratinib could improve the *in vivo* metabolic stability.

To Reviewer #2 (Remarks to the Author):

Yanyan Diao et al. are reporting a generative modeling study of developing macrocyclic compounds in this manuscript. Indeed, macrocyclic molecules are playing an important role in the modern drug discovery. Using deep learning to realize the automated generation of macrocyclic compounds is a beneficial addition to the application scope of generative chemistry. Overall the manuscript is well-organized with a detailed description on the method development and prospectively designed experimental validations. However, the reviewer does have some major and minor concerns to point out.

Response: We appreciate the reviewer's positive and helpful comments on our work and manuscript. Accordingly, we have revised our manuscript fully taking into account the reviewer's concerns.

1. The prospective case study on developing JAK2 inhibitor doesn't support the claimed advantages of adapting macrocyclic compounds. Authors claimed that 'Comparing with the linear analogues, macrocycles tend to possess pre-organized restricted conformations and extended contacts with targets, thus potentially achieving improved binding affinities, better selectivities and superior pharmacological characteristics.' In the enzyme inhibitory assay and the cellular antiproliferative assay as shown in table 3, newly designed cyclic compounds 1-3 do not exhibit the improvement to the control of Fedratinib. While in the following of kinase selectivity test and the in vivo activity test, the control of Fedratinib is simply missing. Thus, from the presented data, there is no evidence of improved binding affinities, better selectivities, and superior pharmacological characteristics.

Response: According to the reviewer's constructive comments, Fedratinib was tested in the kinase selectivity and the in vivo efficacy evaluation experiments. Besides, the preliminary in vivo pharmacokinetic properties of compound **3** and Fedratinib in mice following intravenous (iv, 5 mg/kg) and oral (po, 5 mg/kg) administration were also investigated. The details can be found in the revised manuscript, and the results are summarized as follows.

- 1) Kinase selectivity** Compound **3** had a lower selectivity score (0.042 vs 0.084) than Fedratinib, hence it showed superior kinome selectivity profile. Remarkably, compound **3** mainly targets the TK group and has negligible effect on CMGC, CAMK, and AGC group, while Fedratinib shows binding to a wide range of kinases.
- 2) Pharmacokinetic analysis** Compound **3** displayed overall superior PK properties than Fedratinib. After oral dosing, compound **3** showed longer half-life ($T_{1/2}$, 10.07 vs 4.70 h) and higher systemic exposure (AUC_{inf} , 114.69 vs 50.19 h*ng/mL).

3) ***In vivo* activity** Both compound **3** and Fedratinib is are able to ameliorate the symptoms in DSS-induced murine colitis, but compound **3** (5 mg/kg) showed comparable therapeutic efficacy to Fedratinib (10 mg/kg) at a lower dose.

Overall, the improved selectivity and superior pharmacological characteristics of compound **3** have been demonstrated in the revised manuscript. Generally speaking, some but not all the properties in term of binding affinities, selectivities, and pharmacological characteristics will be improved after macrocyclization. Therefore, the description has been modified to a more precise statement in the revised manuscript as “thus potentially achieving improved binding affinities, better selectivities **or** superior pharmacological characteristics”

2. Then the value of the presented deep learning macrocyclic modification in this manuscript shrinks to a method that can bypass intellectual property restrictions of known active molecules. In this scenario, the academic significance is compromised but the significance to the industrial applications still remains. Here comes the second major concern. When developing JAK2 inhibitors, what is the rationale behind transferring Fedratinib into a macrocyclic compound? The ligand structural and protein structural foundations are missing. Or is bypassing the intellectual restrictions of Fedratinib an objective?

Response: Apart from obtaining novel scaffolds bypassing the intellectual restrictions of Fedratinib, the main reason for transferring Fedratinib into a macrocyclic compound is the disappointing kinome selectivity of Fedratinib. Although Fedratinib is reported to be highly selective to JAK2 over other JAK kinase, its selective profile against the wider kinome is barely satisfactory and even worse than the pan-JAK inhibitor Ruxolitinib. As reported by Davis et al.², Fedratinib inhibited 69 wide-type kinases with K_d values lower than 300 nM, while Ruxolitinib only inhibit 31 wide-type kinases. The off-target effects of Fedratinib may cause undesirable side effects. We envisioned that macrocyclization would obtain proprietary skeletons with improved kinase selectivity

and other properties. The rationale with respect to the macrocyclization of Fedratinib has been explained in the revised manuscript.

The following are more specific points.

3. In the Introduction, authors claimed 'Due to the distinctive features, macrocycles are regarded as a privileged chemotype for targeting challenging proteins that are hardly tractable by traditional small molecule drugs, bridging the gap between small molecules and large biologics'. Generally, the reviewer agrees on that. But not all challenging proteins can be targeted by macrocycles. A brief but explicit discussion can be beneficial.

Response: We thank the reviewer's kind advice. The sentence has been modified to a more precise statement as "... macrocycles are regarded as a privileged chemotype for targeting **some** challenging proteins ..." in the revised manuscript. Additionally, we added some specific challenging proteins that can be targeted by macrocycles as examples "For example, macrocycles predominate the marketed inhibitors of hepatitis C virus (HCV) NS3/4A, the shallow and solvent-exposed groove of which is difficult to harbor small molecules. The advantages of macrocycles have also been reported in modulating protein-protein interactions (PPIs) with large flat and dynamic surfaces".

4. In the section of Model overview. 'the number of macro ring is inferior to 1'. What is the definition here of a macro ring? Is it by counting the number of heavy atoms?

Response: Consistent with the definition of a macrocycle, a macro ring is the ring structure containing 12 or more atoms. We have added the statement in the Model overview section of the revised manuscript.

5. Models were trained with different augmentation levels. The method and the strategy of selecting starting atoms and the direction of writing that structure into

the SMILES string should be specified. The augmentation is not enumerating all possibilities but is defined with 2X, 5X, and 10X.

Response: The method used for data augmentation using substructure-based randomized SMILES has been added to the Data Augmentation section of the Methods parts in the revised manuscript. The process was implemented in RDKit. By setting the doRandom parameter of the MolToSmiles function as True, RDKit would randomly select a starting node and the topological path to enumerate the linear molecule graph. In fact, although the atom indexes of macrocycles were re-ordered according to that of the acyclic substructure by entering the list we pre-arranged, RDKit would alter atom order on-the-fly to prevent strange combinations. This ensures that more reasonable SMILES are generated on the basis of aligning the input and output strings as much as possible.

The number of random SMILES strings for each macrocyclic compound varies widely. If enumerating all possibilities, it will cause a serious imbalance in the data set. After referring to the work of Moret et al.³, the augmentation is defined with 2X, 5X, and 10X. As shown in our manuscript, the 5-fold augmentation has shown good performance in terms of all metrics, and 10-fold data augmentations didn't result in further significant improvement.

6. As authors pointed out that data augmentation helps in learning the basic syntax of the chemical language to produce chemically meaningful SMILES strings. Could a pre-trained model be produced first by learning general ChEMBL / ZINC molecules? Then the transfer learning process can be performed to add specificity to the macrocyclic molecules.

Response: We tested the proposal of transfer learning, as described by the reviewer, using a pre-trained model with general ChEMBL molecules. The general ChEMBL dataset used in this experiment was the one reported in Yang et al.'s work⁴. The model was trained for 100000 steps using general ChEMBL molecules, and the training continued using the macrocyclic ChEMBL training dataset for 100000 steps until the

model converged very well. No augmentation was applied in the transfer learning process. As shown in Table R2 and R3, the transfer learning method improved the uniqueness metric, but the recovery, validity, and linker novelty ($\text{Novelty}_{\text{linker}}$) values greatly decreased. Overall, we think that the transfer learning method couldn't completely replace our data augmentation strategy.

Table R2. Comparison of Macformer with different augmentation numbers and transfer learning method on ChEMBL test dataset.

Method	aug.	Recovery (%)	Validity (%)	Uniqueness (%)	Novelty _{mol} (%)	Novelty _{linker} (%)	Macro-cyclization (%)
Macformer	None	2.70±1.31	72.91±2.05	47.74±8.98	96.10±0.81	44.24±2.05	96.39±0.71
	×2	76.37±3.23	81.97±1.20	44.99±5.37	99.31±0.19	53.03±0.65	99.48±0.08
	×5	81.86±0.75	84.73±1.01	45.14±4.60	99.39±0.09	53.98±1.00	99.53±0.05
	×10	84.25±0.84	85.35±1.33	45.26±0.46	99.43±0.09	50.00±0.95	99.27±0.07
Transfer learning	/	18.21±23.53	58.65±12.40	64.07±16.87	94.46±9.10	38.34±3.28	97.03±0.40

Table R3. Comparison of Macformer with different augmentation numbers and transfer learning method on ZINC test dataset.

Method	aug.	Recovery (%)	Validity (%)	Uniqueness (%)	Novelty _{mol} (%)	Novelty _{linker} (%)	Macro-cyclization (%)
Macformer	None	2.70±1.31	72.91±2.05	47.74±8.98	96.10±0.81	44.24±2.05	96.39±0.71
	×2	76.37±3.23	81.97±1.20	44.99±5.37	99.31±0.19	53.03±0.65	99.48±0.08
	×5	81.86±0.75	84.73±1.01	45.14±4.60	99.39±0.09	53.98±1.00	99.53±0.05
	×10	84.25±0.84	85.35±1.33	45.26±0.46	99.43±0.09	50.00±0.95	99.27±0.07
Transfer learning	/	10.87±19.32	60.82±11.36	65.14±18.16	99.66±0.49	38.32±2.01	98.13±0.40

7. Zinc macrocycles are having lower M.W. than ChEMBL macrocycles. But it is inappropriate of claiming them to be ‘more drug-like’. M.W. between 500--800 can hardly be considered as ‘drug-like’ molecules. The length of SMILES strings

is usually correlates to the M.W. But we don't consider the length of SMILES strings as a 'drug-like property'.

Response: We thank the review for pointing out the inappropriate statement, which has been deleted in the revised manuscript.

8. In Fig.2, t-SNE plots was reported for generated and raw ChEMBL compounds. Adding generated ZINC macrocyclic compounds can be interesting to compare as those compounds were never seen during the training.

Response: In the revised manuscript, we added a side-by-side comparison between our deep learning method and the traditional non-deep learning method as suggested by Reviewer #3. Therefore, in order to more clearly show the results of different methods on different data sets, the t-SNE plots of molecular properties was replaced by the violin plots of each molecular property (Fig S4 for ChEMBL test dataset and Fig S5 for ZINC test dataset). In addition, as suggested by the reviewer, a corresponding discussion of newly generated ZINC macrocycles has been added throughout the "Properties of generated novel macrocycles" section of the revised manuscript.

9. t-SNE plots also suggested that the generation can never jump out of the known chemical space. In the Introduction authors commented that Wagner et al. and Sindhikara et al.'s methods are limited to the collection of fragment linkers. But it is the same story with authors' method here, the creativity of the generation is limited to the original training set.

Response: To actually compare our model with previously reported non-deep learning approaches, we proposed a pipeline to construct macrocycles from 3D structures of linear compounds through linker database searching (termed as MacLS), following the works of Wagner et al. and Sindhikara et al. in the revised manuscript. After comparing the chemical space of generated novel macrocycles by the two methods, we found that the chemical space of the Macformer generated novel macrocycles are closer to that of the real bioactive ones in ChEMBL test dataset. For the ZINC test dataset that are never

seen during the training, the result is pretty much the same. We consider this phenomenon as an advantage of Macformer, since the macrocycles in internal ChEMBL and external ZINC test datasets are real and bioactive in nature. In addition, the potential of the MacLS in macrocyclization of Fedratinib was also evaluated, and the result exhibited that macrocycles generated by Macformer have lower predicted docking scores (the lower the better) than those by MacLS. The advantages of Macformer should be attributed to its Transformer architecture and the data augmentation strategy, which help the model understand the relationship between the input acyclic and target macrocyclic SMILES and add proper linker compatible to the given acyclic structure to generate suitable macrocycles. On the contrary, MacLS cannot learn the prior knowledge about the structural information of macrocyclic compounds through the training process like Macformer, and only considering the local structural information associated with the connected atoms is not conducive to the understanding of the overall macrocyclic structure.

Based on the systematic comparison of the two types of methods, the description of the limitations of previously reported method has been modified to “However, these methods can only enumerate pre-built linker libraries, without the ability to derive new linkers of structural novelty. The only consideration of the local structural information associated with the connected atoms is also not conducive to the understanding of the overall macrocyclic structure.” in the introduction part of the revised manuscript.

To Reviewer #3 (Remarks to the Author):

This manuscript describes the development and application of a computational method termed "Macformer" to propose chemical linkers suited to convert acyclic ligands into macrocyclic ones. The authors indicate that the software tool takes advantage of deep learning principles. They further indicate that they have

trained the software and applied it then to propose cyclization linkers of the approved drug fedratinib being an acyclic small molecule JAK2 inhibitor. The software tool proposed 281 macrocyclic compounds of which they synthesized three and tested them in vitro. One was highly potent, showed high target selectivity and proved active in vivo.

Reading the title and abstract, I was excited to read the study, but I got disappointed by the work the more I read it in detail. In particular, I was shocked when I found data in literature in which the linker that the authors "found" by the new software tool was already reported, and is present in an approved macrocyclic JAK inhibitor drug (pacritinib). I describe my four major concerns in the following, all being alone a reason to decline the work. I do not recommend publication.

Response: We thank the reviewers for pointing out the shortcomings of our study, which is of great help to improve the quality of our paper. Accordingly, we have revised our manuscript fully taking into account the reviewer's concerns.

1. Difficulty to understand computational methods and concepts behind:

I found it hard to understand the computational methods described. The concepts are not well described and are rather cryptic. There are many examples of studies that describe complex computational methods but manage to describe the concepts in an understandable way (e.g. the recent Cell paper by the Baker lab which I read and enjoyed as it clearly describes the concept(s) of the new approach). Without making the concepts/principles understandable for the medchem community, this work is of little value, and for reviewers like me, it is not possible to evaluate the concepts behind the software tool.

Response: According to the reviewer's critical comments, we have modified our manuscript. While exhibiting the necessary experimental details of the computational method to help researchers replicate our results, we added some descriptions about the concept of behind our computational method and why we chose this method. We hope

the revised manuscript will help more researchers in related fields understand the new method, and as pointed out by the review, it is indeed crucial for exhibiting the value of our method.

2. Taking advantage of structural information:

It appears to me that the software tool does not take advantage of the structural information to propose linker structures suited to cyclize acyclic ligands. In particular for the applied example, there is an X-ray structure of JAK2 with Fedratinib bound, which can be taken to measure the distance between the two points that need to be connected, as well as their orientation. A computational tool that does not use this information will never be superior to a conventional approach where structure information is taken into account and potential linkers are modeled.

Response: Our deep learning model Macformer is based on the Transformer architecture, which is the state-of-the-art neural network model to deal with sequential data and has been used in many drug design tasks, such as de novo molecule generation, and retrosynthesis planning. During the training process of Macformer, both acyclic source and macrocyclic target SMILES will be fed to the model, then the model learn how to generate macrocycles from the input SMILES of acyclic structures. When the output SMILES were inferred, Macformer processed the input SMILES in a holistic manner. This means that all tokens in the source acyclic SMILES strings, albeit with different significance, will participate in the generation of the macrocyclic SMILES strings in Macformer. Furthermore, we adopted data augmentation using substructure-aligned randomized SMILES, which exposed the structural information of the acyclic and macrocyclic SMILES and the relationship between them to the model from various views. Although no explicit 3D structural information was provided, Macformer may learn such knowledge implicitly from the SMILES sequence of ChEMBL bioactive macrocycles (the training dataset) and add proper linker compatible to the given acyclic structure to generate suitable macrocycles.

The traditional structure-based macrocyclization method, as described by the reviewer, consider the matching of geometric parameters related to the formation of new macrocyclic chemical bonds, in order to maintain the known bioactive orientation of the linear compound. This very direct approach does provide intuitive information about the 3D structure of macrocyclic compounds, and it has been successfully used in the design of many macrocyclic lead compounds or even drugs. However, we think that only considering the local structural information associated with the connected atoms is not conducive to the understanding of the overall macrocyclic structure. Generally, during the overall structural optimization or target-induced binding of the macrocyclized compounds, the conformation of the acyclic substructure may be changed to deviate from the active one. Therefore, if constructing macrocycles in an alternative global and non-local conformational strictly constrained way, we may identify novel potent lead compounds that might have been missed by traditional approaches.

In the revised manuscript, we compared our deep learning method with the traditional method. The issue will be discussed in detail in the next question.

3. Comparison with conventional methods:

The authors do not side-by-side compare their tool with conventional methods. Reading the literature, I found that several groups and companies have already macrocyclized acyclic ligands of JAK kinases, and a comparison would be required. For the reasons described in point 2, it is essentially impossible that the developed tool can be better than conventional methods that take advantage of available structure information.

Response: Thanks for the reviewer's constructive comments. According to the reviewer's suggestion, we have compared our deep learning method with the traditional non-deep learning method in a side-by-side manner. Although the traditional computational method regarding automatic macrocyclization has been mentioned in many drug design cases, there are no publicly available tools. In the revised manuscript,

we proposed a pipeline to construct macrocycles from 3D structures of linear compounds through linker database searching (termed as MacLS), following the previously reported works. To ensure that macrocycles are constructed using the same data, linkers extracted from the ChEMBL training dataset were used to construct the 3D linker database. The detailed experimental process and results can be found in the revised manuscript, and the results are summarized as follows.

1) Evaluation on internal ChEMBL and external ZINC test datasets (10 macrocyclic candidates were output for each linear structure)

- The macrocycles generated by Macformer have lower uniqueness and molecular novelty. (disadvantage)
- Macformer could derive new linkers of structural novelty that are not present in the training dataset, whereas MacLS can only enumerate the linker libraries constructed using linkers of the training dataset. (advantage)
- Macformer could reconstruct the target macrocycles in the test datasets with significant higher ratio than MacLS (84.25~97.54 % vs 4.16-4.52 %). (advantage)
- The novel macrocycles generated by Macformer have higher structural similarity and closer molecular properties to the target macrocyclic compounds than MacLS. The results indicate that the chemical space of the Macformer generated novel macrocycles are closer to that of the real bioactive ones, implying the potential of Macformer in practical drug design process. (advantage)

2) Performance for macrocyclic JAK2 inhibitor design (281 and 300 macrocycles generated by Macformer and MacLS, respectively)

- The macrocycles generated by Macformer have lower (the lower the better) predicted docking scores than those by MacLS.
- None of the three compounds selected for further experimental validation is present in the macrocycles obtained through MacLS.

In summary, we think the above results have demonstrate the applicability of our method in identifying novel potent lead compounds that the traditional method might have missed. It is expected that, as a new method that construct macrocycles from a

different view and a powerful complement to the traditional macrocyclization method, Macformer will play a valuable role in the design of macrocyclic drug candidates.

4. The "identified" linker in compound 3 is already reported in literature:

The linker of the best one of the three compounds tested is already reported in literature. It is present in the approved drug Pacritinib, which is a macrocyclic JAK inhibitor (!). The authors do not mention this inhibitor in their report and neither describe relevant efforts to macrocyclize acyclic JAK inhibitors by other methods. An audience seeing the results of this work would immediately think that the authors have been cheating and simply have stolen the linker from the approved drug. They would not believe that it was truly identified by the new software tool. Personally, I think that presenting the data as done herein, without mentioning that this linker was found in other macrocyclic ligands of JAK targets (and even in an approached drug), is not honest and poor scientific practice.

Response: Despite of the structural novelty of compound 3, its linker does appear in many existing macrocycles, especially in the approved JAK2 inhibitor Pacritinib. According to the reviewer's kind advice, we have clarified this issue in the revised manuscript.

To Reviewer #4 (Remarks to the Author):

The mechanisms used for drug design and selection look interesting. The colitis part in this manuscript is weak, however. First, I think that Jak2 is certainly not considered as a good target for gut inflammation in humans given the issues on myelotoxicity and thus studies in humans currently focus on Jak1 and to a lesser extent on Tyk2. Second the mouse model used is a model of acute intestinal injury

and does not mimick chronic inflammation in humans. Finally, the authors showed prevention studies but did not demonstrate the therapeutic value of their compound.

Response: We thank the reviewer for the positive assessment on our strategy used for macrocyclic drug candidate design. Regarding the three issues of the colitis model that the reviewer is concerned about, we will elaborate one by one in the following sections.

1. First, I think that Jak2 is certainly not considered as a good target for gut inflammation in humans given the issues on myelotoxicity and thus studies in humans currently focus on Jak1 and to a lesser extent on Tyk2.

Response: The import roles of JAK2 in the pathogenesis of inflammatory bowel disease (IBD) have been reported by many studies. For example, the study of Song et al.⁵ using intestinal mucosal biopsies has shown that the mRNA and protein expression of JAK2 were significantly increased in Chinese patients with ulcerative colitis (UC) and Crohn's disease (CD), two subtypes of IBD. In the study conducted by Asadzadeh-Aghdaei et al., 246 patients with IBD were enrolled, and JAK2 mRNA upregulation was also observed⁶. These findings suggest the potential of JAK2 as a therapeutic target for IBD. Although JAK2 inhibitors have not been reported in clinical research and marketed drugs related to IBD, the investigational research continues. For example, Rivera revealed the overall protective role for intestinal epithelial JAK2 in intestinal homeostasis⁷. Besides, many traditional Chinese herbs have been reported to ameliorate DSS-induced colitis in mice via suppressing JAK2/STAT3 signaling^{8,9}.

The exclusion of JAK2 from IBD treatment is mainly due to its engagement in signaling of hematopoietic cytokines, which may cause the undesirable side effect of myelosuppression. Ruxolitinib is a selective JAK2/JAK1 inhibitor, with IC₅₀ values of 2.8 and 3.3 nM, respectively. Ruxolitinib has been approved for treatment of myelofibrosis and polycythemia vera (PV), but it carries the undesirable side effect of myelosuppression. Swee and coworkers reported the successful use of Ruxolitinib to treat a patient with concomitant UC and PV¹⁰. The specific case sheds light on

alternative therapies for UC treatment from a special perspective and underscores the importance to explore the broad efficacy and safety of selective JAK inhibitors.

In summary, continued efforts are needed to clarify the potential therapeutic efficacy of JAK2 inhibitors in IBD. We hope that our preliminary exploration can play a role in attracting new ideas.

2. Second the mouse model used is a model of acute intestinal injury and does not mimick chronic inflammation in humans.

Response: Human IBD is a chronic, relapsing inflammatory disorder of the gastrointestinal tract, its etiology and pathogenesis are complicated and still uncertain. Due to its rapidity, simplicity, and reproducibility, the dextran sulfate sodium (DSS)-induced colitis model is widely used in IBD research as a preclinical model. Although there are differences between the acute DSS colitis model in our study and human IBD, they share many clinical and pathological features, such as bloody stools, weight loss, diarrhea, and inflammatory cells infiltration. The acute colitis model has been used for studying the pathogenesis of IBD as well as evaluating the efficacy of many drug candidates including those involved in JAK/STAT signaling¹¹⁻¹³.

3. Finally, the authors showed prevention studies but did not demonstrate the therapeutic value of their compound.

Response: As described in the In vivo Efficacy Study section of the Methods part of the manuscript, Male BALB/c mice were given 3.5% DSS water daily for 7 days to induce colitis, and the tested compounds were administrated from day 8. As shown in Fig. 7, compound **3** could alleviate the decrease of body weight, reduce the disease activity index (DAI) score, and improve the colonic histological changes, demonstrating the therapeutic value of our compound.

References

1. Werning, G. et al. Efficacy of TG101348, a selective JAK2 inhibitor, in

- treatment of a murine model of JAK2V617F-induced polycythemia vera. *Cancer Cell* **13**, 311-320 (2008).
2. Davis, M. I. et al. Comprehensive analysis of kinase inhibitor selectivity. *Nat Biotechnol* **29**, 1046-U1124 (2011).
 3. Moret, M., Friedrich, L., Grisoni, F., Merk, D. & Schneider, G. Generative molecular design in low data regimes. *Nat Mach Intell* **2**, 171-180 (2020).
 4. Yang, Y. Y. et al. SyntaLinker: automatic fragment linking with deep conditional transformer neural networks. *Chem Sci* **11**, 8312-8322 (2020).
 5. Song, L. et al. High Intestinal and Systemic Levels of Interleukin-23/T-Helper 17 Pathway in Chinese Patients with Inflammatory Bowel Disease. *Mediat Inflamm* **2013** (2013).
 6. Asadzadeh-Aghdaei, H. et al. V617F-independent upregulation of JAK2 gene expression in patients with inflammatory bowel disease. *J Cell Biochem* **120**, 15746-15755 (2019).
 7. Rivera, D. C. Investigating the Role of JAK2 in Intestinal Homeostasis and Disease. *A thesis submitted in conformity with the requirements for the degree of Master of Science, Pharmacology and Toxicology, University of Toronto* (2022).
 8. Zhao, Y. X. et al. Gegen Qinlian decoction relieved DSS-induced ulcerative colitis in mice by modulating Th17/Treg cell homeostasis via suppressing IL-6/JAK2/STAT3 signaling. *Phytomedicine* **84** (2021).
 9. Lu, Z. et al. Huanglian Jiedu Decoction ameliorates DSS-induced colitis in mice via the JAK2/STAT3 signalling pathway. *Chin Med-Uk* **15** (2020).
 10. Swei, E. C., Fox, C. M., Bowles, D. W., Rizeq, M. N. & Onyiah, J. C. Use of Ruxolitinib for the Simultaneous Treatment of Severe Refractory Ulcerative Colitis and Polycythemia Vera. *ACG Case Rep J* **9**, e00741 (2022).
 11. Spalinger, M. R. et al. The JAK Inhibitor Tofacitinib Rescues Intestinal Barrier Defects Caused by Disrupted Epithelial-macrophage Interactions. *J Crohns Colitis* **15**, 471-484 (2021).
 12. Li, Y., Altemus, J. & Lightner, A. L. Mesenchymal stem cells and acellular products attenuate murine induced colitis. *Stem Cell Res Ther* **11** (2020).
 13. Sann, H., Erichsen, J., Hessmann, M., Pahl, A. & Hoffmeyer, A. Efficacy of drugs used in the treatment of IBD and combinations thereof in acute DSS-induced colitis in mice. *Life Sci* **92**, 708-718 (2013).

REVIEWER COMMENTS

Reviewer #1 (Remarks to the Author):

The authors have addressed my concerns.

Reviewer #2 (Remarks to the Author):

Authors have successfully addressed questions and comments the reviewer had. The reviewer considers the revision to be complete and thorough.

Reviewer #3 (Remarks to the Author):

The authors have discussed extensively the questions raised and they have added text/made larger changes, but they have not addressed my concerns that I repeat below. I think that this work does not have the required relevance and quality to be published, neither in Nature Communications, not in a more specialized journal.

1. Conceptual problem with approach and poor description/rationalization of concepts:

It remains still unclear to me why the proposed strategy/workflow should be able to predict optimal linkers that turn linear ligands into good macrocyclic ligands. The predictions are made based on a tool that is trained with a large number of random macrocycles binding random targets. The structure of the target is not entered into the equation. I thus think that the "macformer" tool cannot propose a linker that leads to optimal binding to a specific target (because the tool does not know the target!). If the authors think that this is nevertheless possible, they should demonstrate this with a compelling example (which they did not).

2. Prediction without structural information:

By all means, I cannot understand how a linker can be proposed if the structural context is not taken into account. An optimal linker depends absolutely on the structure of the target protein, as the

trajectory it needs must to not clash with the protein surface and at the same time should not be too distant so that it can pick up contacts with the protein to increase binding affinity.

3. Comparison to conventional methods:

In the revised manuscript, the authors describe an effort that they have made. However, the description is hard to follow. The parts that I can understand are not at all convincing to me.

4. The "identified" linker was already reported:

As described before, the described linker was found before. The authors declare this now. However, a reader would still think that the authors have simply been cheating by taking a linker that proofed good before. If the "macformer" tool is truly working (and I have doubts that it does), the authors would need to apply it to other targets and come up with linkers that were not reported for the same target. And they should show that the developed macrocycles have truly better properties like binding affinity or specificity.

Reviewer #4 (Remarks to the Author):

To Reviewer #4 (Remarks to the Author):

The mechanisms used for drug design and selection look interesting. The colitis part in this manuscript is weak, however. First, I think that Jak2 is certainly not considered as a good target for gut inflammation in humans given the issues on myelotoxicity and thus studies in humans currently focus on Jak1 and to a lesser extent on Tyk2. Second the mouse model used is a model of acute intestinal injury and does not mimick chronic inflammation in humans. Finally, the authors showed prevention studies but did not demonstrate the therapeutic value of their

compound.

Response: We thank the reviewer for the positive assessment on our strategy used for macrocyclic drug candidate design. Regarding the three issues of the colitis model that the reviewer is concerned about, we will elaborate one by one in the following sections.

1. First, I think that Jak2 is certainly not considered as a good target for gut inflammation in humans given the issues on myelotoxicity and thus studies in humans currently focus on Jak1 and to a lesser extent on Tyk2.

Response: The important roles of JAK2 in the pathogenesis of inflammatory bowel disease (IBD) have been reported by many studies. For example, the study of Song et al.⁵ using intestinal mucosal biopsies has shown that the mRNA and protein expression of JAK2 were significantly increased in Chinese patients with ulcerative colitis (UC) and Crohn's disease (CD), two subtypes of IBD. In the study conducted by Asadzadeh Aghdai et al., 246 patients with IBD were enrolled, and JAK2 mRNA upregulation was also observed⁶. These findings suggest the potential of JAK2 as a therapeutic target for IBD. Although JAK2 inhibitors have not been reported in clinical research and marketed drugs related to IBD, the investigational research continues. For example, Rivera revealed the overall protective role for intestinal epithelial JAK2 in intestinal homeostasis⁷. Besides, many traditional Chinese herbs have been reported to ameliorate DSS-induced colitis in mice via suppressing JAK2/STAT3 signaling^{8,9}.

The exclusion of JAK2 from IBD treatment is mainly due to its engagement in signaling of hematopoietic cytokines, which may cause the undesirable side effect of myelosuppression. Ruxolitinib is a selective JAK2/JAK1 inhibitor, with IC₅₀ values of 2.8 and 3.3 nM, respectively. Ruxolitinib has been approved for treatment of myelofibrosis and polycythemia vera (PV), but it carries the undesirable side effect of myelosuppression. Swei and coworkers reported the successful use of Ruxolitinib to treat a patient with concomitant UC and PV¹⁰. The specific case sheds light on alternative therapies for UC treatment from a special perspective and underscores the importance to explore the broad efficacy and safety of selective JAK inhibitors.

In summary, continued efforts are needed to clarify the potential therapeutic efficacy

of JAK2 inhibitors in IBD. We hope that our preliminary exploration can play a role in attracting new ideas.

REPLY TO AUTHORS: Sorry but this is not convincing. Simply citing some papers on JAK2 expression does not implicate that this is a good target. Nobody in the IBD field would consider JAK2 as a good target (in contrast to JAK1). The authors have not addressed this point.

2. Second the mouse model used is a model of acute intestinal injury and does not mimick chronic inflammation in humans.

Response: Human IBD is a chronic, relapsing inflammatory disorder of the gastrointestinal tract, its etiology and pathogenesis are complicated and still uncertain. Due to its rapidity, simplicity, and reproducibility, the dextran sulfate sodium (DSS)-induced colitis model is widely used in IBD research as a preclinical model. Although there are differences between the acute DSS colitis model in our study and human IBD, they share many clinical and pathological features, such as bloody stools, weight loss, diarrhea, and inflammatory cells infiltration. The acute colitis model has been used for studying the pathogenesis of IBD as well as evaluating the efficacy of many drug candidates including those involved in JAK/STAT signaling¹¹⁻¹³.

REPLY TO AUTHORS: Sorry but this is not convincing. The authors have done nothing to address my point.

3. Finally, the authors showed prevention studies but did not demonstrate the therapeutic value of their compound.

Response: As described in the In vivo Efficacy Study section of the Methods part of the manuscript, Male BALB/c mice were given 3.5% DSS water daily for 7 days to induce colitis, and the tested compounds were administrated from day 8. As shown in Fig. 7, compound 3 could alleviate the decrease of body weight, reduce the disease activity index (DAI) score, and improve the colonic histological changes,

demonstrating the therapeutic value of our compoun

REPLY TO AUTHORS: Sorry but this is not convincing. Data in chronic colitis models are needed.

To Reviewer #3 (Remarks to the Author):

The authors have discussed extensively the questions raised and they have added text/made larger changes, but they have not addressed my concerns that I repeat below. I think that this work does not have the required relevance and quality to be published, neither in Nature Communications, not in a more specialized journal.

Response: We politely disagree with the reviewer's comments on our work. We think that the differences in areas of expertise and background knowledge prevent the reviewer from fully understanding our work. Accordingly, to better illustrate our work and address the reviewer's concerns, we would like to discuss some conceptual issues with respect to the macrocyclization task first.

The first three questions of the reviewer mainly focus on the absent of target information in Macformer when inferring the linkers and the comparison with conventional methods. So we would like to start by reviewing the reported macrocyclization work.

The reported successful rational design of macrocycles against a specific target are commonly summarized as "structure-based". Here, the "structure" actually contains two aspects, the structure of the acyclic/macrocyclic compound and the structure of the target. This process can usually be divided into two steps, the addition of macrocyclic linkers that are compatible with the linear compound to form macrocycles and the evaluation of the compatibility between the macrocycles and the binding pocket of the target. Although sometimes the boundary between the two steps is blurred. For the second step, available research methods are relatively explicit, and many simulation methods commonly used in drug design, such as conformation optimization and molecular docking, can assist this process. Undoubtedly, as a prerequisite and guarantee to obtain macrocyclic candidates, it is critical to add appropriate linkers that are compatible with the starting acyclic compounds.

However, the macrocyclization of linear compounds in the initial stage is driven primarily by the empirical knowledge of the medicinal chemists, where the final results are presented directly, while the detailed process is less described in the literatures. This

opaque and non-standardized procedure is difficult for inexperienced researchers to follow, and the empirical knowledge is insufficient to cover the vast chemical space of the macrocyclic linkers. Computational tools that utilized geometrically constrained linker database searching and linker connection strategy to generate macrocycles from the 3D structure of a given acyclic ligand have also been reported. The computational methods are similar to the reviewer's comments in the second question of the first round of review, that is *"there is an X-ray structure of JAK2 with Fedratinib bound, which can be taken to measure the distance between the two points that need to be connected, as well as their orientation."* However, the reported computational methods has some drawbacks. First, it can only enumerate pre-built linker libraries, without the ability to derive new linkers of structural novelty. Second, the only consideration of the local conformational matching associated with the connected atoms is also not conducive to the understanding of the overall macrocyclic structure.

The deep learning based Macformer method proposed here is designed to solve the problem of adding proper linkers compatible with the acyclic compound to generate macrocycles in the initial stage. It is a deep generative model dedicated to generate diverse and novel macrocyclic analogues of the given acyclic molecules. Due to the absence of explicit targets for many bioactive macrocycles, the target information was not involved in Macformer. Therefore, in the JAK2 macrocyclic inhibitor design, the macrocycles generated by Macformer were docked into the ATP binding site of JAK2 to further evaluate their interactions with the target, which were used as an import criteria for subsequent compound selection.

In the revised manuscript, we have clarified the function of Macformer in the design paradigm of the macrocyclic drug candidates. Meanwhile, the reliance of Macformer on methods that assess the macrocycle-target interactions, such as molecular docking, is highlighted in the revised manuscript.

1. Conceptual problem with approach and poor description/rationalization of concepts: It remains still unclear to me why the proposed strategy/workflow should be able to predict optimal linkers that turn linear ligands into good

macrocyclic ligands. The predictions are made based on a tool that is trained with a large number of random macrocycles binding random targets. The structure of the target is not entered into the equation. I thus think that the "macformer" tool cannot propose a linker that leads to optimal binding to a specific target (because the tool does not know the target!). If the authors think that this is nevertheless possible, they should demonstrate this with a compelling example (which they did not).

Response: As pointed out by the reviewer, the target information was not involved in Macformer. Given an acyclic molecule, the purposes of Macformer are to generate diverse and novel macrocyclic analogues before further evaluation of binding potential against the target of interest. Accordingly, we cannot guarantee that all the linkers proposed by Macformer could lead to optimal binding to a specific target. In the prospective design of novel macrocyclic JAK2 inhibitors, we used molecular docking to evaluate the interactions with the target and filter the macrocycles generated by Macformer. In fact, as illustrated in Fig. S7 of the supplementary information, many of the macrocycles generated by Macfomer have poor docking scores, suggesting poor binding to JAK2.

2. Prediction without structural information:

By all means, I cannot understand how a linker can be proposed if the structural context is not taken into account. An optimal linker depends absolutely on the structure of the target protein, as the trajectory it needs must to not clash with the protein surface and at the same time should not be too distant so that it can pick up contacts with the protein to increase binding affinity.

Response: The structure of the target protein is certainly important for the design of appropriate macrocyclic linkers. But, before that, the linkers need to be compatible with the structure of the starting acyclic compound. In Macformer, the Transformer architecture was used to grasp the whole structural information of the acyclic and macrocyclic compounds through global modeling on SMILES sequences and infer

more suitable macrocyclic linkers compatible with the given acyclic molecules. We think that the overall compatibility of the starting acyclic compounds with the linkers may help preserve the activities of the acyclic compounds and facilitate the binding of the macrocycles with the target.

(This paragraph has little to do with Macformer and is mainly for the purpose of academic discussion. In our opinion, avoiding clashes with the protein surface is the primary concerns when adding macrocyclic linkers, while the long distant to the target is not always a disadvantage. The linkers that could maintain the critical interactions of the starting linear compound with the target, regardless of whether they have contacts with the target, are favorable to increase binding affinity by delivering entropic benefit to potency through the rigid conformation of the macrocycles.)

3. Comparison to conventional methods:

In the revised manuscript, the authors describe an effort that they have made. However, the description is hard to follow. The parts that I can understand are not at all convincing to me.

Response: To the best of our knowledge, the majority of the reported works associated with rational design of macrocyclic drug candidates are based on the experts' experience and knowledge of medicinal chemistry. In these works, it is usually a lengthy presentation of the discovery of a specific macrocyclic drug candidate, without involving comparisons with other methods. Hence, it is a thorny issue to define what the conventional methods are.

According to the discussion we put forward at the beginning of the response letter, the function of Macformer in macrocyclic drug candidates design is to explore the vast chemical space of the macrocyclic analogues of a given acyclic molecule in the initial stage by adding linkers compatible with the acyclic molecule. And the generated macrocycles need to be filtered and validated through other molecular simulation methods, such as docking. We think it is reasonable to compare Macformer with methods that have the same functions. In the initial stage of macrocyclization, the

selection of linkers are mainly driven by the empirical knowledge of the medicinal chemists, where the final linkers are commonly presented directly after a short description of the purpose while the detailed process is absent. This artificial and non-standardized procedure could not be a baseline for comparison.

Instead, we used the linker database searching computational method as the baseline, which applies geometric criteria, e.g., distance and angle compatibility between the atoms to be connected, to form initial macrocycles based on the three-dimensional (3D) structure of an acyclic ligand (termed as MacLS method in our manuscript). In fact, we think the computational method is consistent to the reviewer's suggestion in the second question of the first round of review, that is "*there is an X-ray structure of JAK2 with Fedratinib bound, which can be taken to measure the distance between the two points that need to be connected, as well as their orientation.*" It should be emphasized that the structure of the target is also not considered in MacLS for a fair comparison. In the revised manuscript, we modified the description about the MacLS method for better understanding. Additionally, the Fig. S2 in the supplementary information provides a visual representation to facilitate understanding.

4. The "identified" linker was already reported:

As described before, the described linker was found before. The authors declare this now. However, a reader would still think that the authors have simply been cheating by taking a linker that proofed good before. If the "macformer" tool is truly working (and I have doubts that it does), the authors would need to apply it to other targets and come up with linkers that were not reported for the same target. And they should show that the developed macrocycles have truly better properties like binding affinity or specificity.

Response: We would like to address the concern of the reviewer from three aspects.

1) The source SMILES file that was used as the input for generation of macrocyclic analogues of Fedratinib has been upload in GitHub (https://github.com/yydiao1025/Macformer/blob/main/data/src_fedratinib.txt). With

the source code of Macformer and the pre-trained models, it is not a difficult task to verify if Macformer could actually generate the three macrocycles we have synthesized. 2) As pointed out by the reviewer in first round of review “*there is an X-ray structure of JAK2 with Fedratinib bound*” (PDB code 6VNE), but if we “*measure the distance between the two points that need to be connected, as well as their orientation*”, the linker of compound 3 would not be considered as an appropriate macrocyclic linker for Fedratinib.

Because of the structural difference between Pacritinib and Fedratinib, the additional -NH- group makes the two terminal phenyl groups of Fedratinib close to each other. The close distance between the two macrocyclization connection points, atoms labeled as a_2 and a_3 in Fig. R1a, makes the long linker with eight atoms of Pacritinib incompatible with Fedratinib. To thoroughly probe the linker, we acquired its conformations from three ways: 1) generate the conformations from its SMILES using RDKit and 26 conformations were obtained (No. 1-26 in Table R1), 2) extract from the model conformations of ligands reported in the PDB dataset and 15 conformations were obtained (No. 27-41 in Table R1), 3) extract from the lowest-energy 3D structures of ChEMBL macrocycles and 45 conformations were obtained (No. 42-86 in Table R1),

For a proper linker, its attachment vectors should match with that of Fedratinib. Herein, an attachment vector is the bond between the atom at the cyclization site and the leaving atom that will not be contained in the generated macrocycles. As shown in Fig. R1a, the attachment vectors for Fedratinib is the bond between atoms a_1 and a_2 , and a_3 and a_4 , respectively. When a macrocycle is formed, the leaving atoms a_1 and a_4 in Fedratinib and atoms a'_2 and a'_3 in the linker will not be contained in the macrocycle. Ideally, the differences between distances ($|d_1 - d'_1|$ and $|d_2 - d'_2|$) and dihedral angles ($|da - da'|$) should be 0. The bigger difference values means the worse compatibility between Fedratinib and the linker. As shown in Fig. R1b and Table R1, among the 86 conformations, none of them satisfies the distance and dihedral angle cutoff (0.5 Å for distance and 20° for the dihedral angle used in MacLS). To sum up, if we only contain the compatibility between the attachment vectors of Fedratinib and the linkers, which follows the conventional structure-based macrocyclization concept that

tries to keep the conformation of the starting acyclic molecule, the linker of compound 3 would not be considered as an appropriate macrocyclic linker for Fedratinib. On the contrary, Macformer abandoned the strict limits on 3D structures of the ligand and deduced the linker of compound 3 that was validated to exhibit improved kinome selectivity and PK properties than Fedratinib.

Fig. R1 a) Illustration of the geometric parameters that were used to evaluate the compatibility between Fedratinib and the macrocyclic linkers. $d_1 = 2.9 \text{ \AA}$, $d_2 = 4.0 \text{ \AA}$, and the dihedral angle (da) between atoms a_1 , a_2 , a_3 , and a_4 in Fedratinib is -56.46° . b) Plot of difference in distances and dihedral angles between the attachment vectors. And da' is the dihedral angle between atoms a'_1 , a'_2 , a'_3 , and a'_4 in the linker.

Table R1. The geometric parameters of different conformations of the linker of compound 3.

No.	d'_1 (Å)	$ d_1 - d'_1 $ (Å)	d'_2 (Å)	$ d_2 - d'_2 $ (Å)	da' (°)	$ da - da' $ (°)
1	8.16	5.26	10.66	6.66	179.99	236.45
2	7.68	4.78	9.35	5.35	44.88	101.34
3	7.68	4.78	9.35	5.35	44.87	101.33
4	7.68	4.78	9.35	5.35	-44.87	11.59
5	7.68	4.78	9.35	5.35	-44.88	11.58
6	7.76	4.86	9.05	5.05	20.82	77.28
7	7.39	4.49	8.71	4.71	-40.65	15.81
8	7.39	4.49	8.71	4.71	40.66	97.12
9	7.69	4.79	9.88	5.88	-180	123.54

10	7.69	4.79	9.88	5.88	-180	123.54
11	7.53	4.63	9.67	5.67	-159.24	102.78
12	7.09	4.19	9.29	5.29	150.42	206.88
13	6.97	4.07	9.97	5.97	0	56.46
14	5.58	2.68	7.37	3.37	10.37	66.83
15	5.58	2.68	7.37	3.37	-10.36	46.1
16	5.58	2.68	7.37	3.37	-10.36	46.1
17	5.58	2.68	7.37	3.37	10.37	66.83
18	3.77	0.87	4.5	0.5	-97.52	41.06
19	3.77	0.87	4.5	0.5	97.52	153.98
20	3.88	0.98	3.73	0.27	0	56.46
21	3.88	0.98	3.73	0.27	0	56.46
22	3.96	1.06	5.28	1.28	92.29	148.75
23	4.3	1.4	5.11	1.11	-93.04	36.58
24	5.03	2.13	6.5	2.5	-96.49	40.03
25	5.83	2.93	7.17	3.17	-56.01	0.45
26	5.11	2.21	5.09	1.09	-69.36	12.9
27	6.79	3.89	8.36	4.36	45.22	101.68
28	7.7	4.8	9.22	5.22	36.4	92.86
29	4.55	1.65	5.65	1.65	-28.18	28.28
30	6.79	3.89	8.76	4.76	-61.81	5.35
31	4.42	1.52	3.61	0.39	-52.78	3.68
32	7.22	4.32	8.87	4.87	69.77	126.23
33	7.99	5.09	9.43	5.43	23.18	79.64
34	7.85	4.95	9.4	5.4	63.22	119.68
35	7.26	4.36	8.1	4.1	60.93	117.39
36	7.68	4.78	8.2	4.2	-30.55	25.91
37	4.98	2.08	7.23	3.23	-100.11	43.65
38	3.2	0.3	3.49	0.51	35.03	91.49
39	7.34	4.44	9.84	5.84	138.66	195.12
40	7.84	4.94	9.22	5.22	4.69	61.15
41	6.71	3.81	7.26	3.26	-89.17	32.71
42	5.33	2.43	5.63	1.63	77.47	133.93
43	5.44	2.54	6.12	2.12	8.23	64.69
44	5.53	2.63	6.14	2.14	29.95	86.41
45	5.55	2.65	6.17	2.17	6.39	62.85
46	5.54	2.64	6.11	2.11	5.78	62.24
47	5.52	2.62	6.13	2.13	-6.89	49.57
48	6.01	3.11	6.04	2.04	82.62	139.08
49	5.73	2.83	6.01	2.01	-49.87	6.59
50	6.36	3.46	6.18	2.18	-88.23	31.77

51	5.51	2.61	6.11	2.11	26.32	82.78
52	5.49	2.59	6.09	2.09	25.23	81.69
53	4.61	1.71	5.58	1.58	45.05	101.51
54	5.54	2.64	6.18	2.18	-7.01	49.45
55	5.62	2.72	6.15	2.15	-26.48	29.98
56	6.38	3.48	6.21	2.21	91.12	147.58
57	6.39	3.49	6.22	2.22	88.66	145.12
58	6.11	3.21	6.1	2.1	-85.43	28.97
59	5.71	2.81	6.15	2.15	61.26	117.72
60	5.47	2.57	6.1	2.1	-29.58	26.88
61	5.25	2.35	5.55	1.55	-35.35	21.11
62	6.05	3.15	6.09	2.09	-82.86	26.4
63	5.9	3	6.06	2.06	37.86	94.32
64	5.61	2.71	6.21	2.21	-6.44	50.02
65	5.69	2.79	6.04	2.04	-68.87	12.41
66	6.02	3.12	6.06	2.06	-81.95	25.49
67	6.33	3.43	6.12	2.12	-91.96	35.5
68	5.71	2.81	6.02	2.02	52.44	108.9
69	6.05	3.15	6.1	2.1	-83.57	27.11
70	4.76	1.86	5.75	1.75	9.07	65.53
71	5.54	2.64	6.2	2.2	-22.65	33.81
72	5.51	2.61	6.16	2.16	-8.86	47.6
73	5.38	2.48	6.07	2.07	30.81	87.27
74	5.68	2.78	5.98	1.98	78.3	134.76
75	6.03	3.13	6.08	2.08	-83.26	26.8
76	5.99	3.09	6.06	2.06	82.91	139.37
77	6.04	3.14	6.07	2.07	-83.59	27.13
78	5.66	2.76	6.24	2.24	-18.2	38.26
79	6.34	3.44	6.17	2.17	89.85	146.31
80	4.75	1.85	5.71	1.71	11.55	68.01
81	6.05	3.15	6.09	2.09	82.8	139.26
82	5.75	2.85	6.12	2.12	-26.62	29.84
83	5.98	3.08	6.05	2.05	82.87	139.33
84	5.71	2.81	6.23	2.23	-56.66	0.2
85	5.56	2.66	6.12	2.12	-26.7	29.76
86	6	3.1	6.04	2.04	82.17	138.63

3) To address the reviewer’s concern about “*come up with linkers that were not reported for the same target*”, in the revised manuscript, we added wet experiments for

the macrocyclic compound **1** generated by Macformer. To the best of our knowledge, the linker of compound **1** is first reported in our study for the design of macrocyclic and selective JAK2 inhibitor. As displayed in Fig. and Table 4, compound **1** showed improved kinome selectivity and PK properties than Fedratinib.

To Reviewer #4 (Remarks to the Author):

The mechanisms used for drug design and selection look interesting. The colitis part in this manuscript is weak, however. First, I think that Jak2 is certainly not considered as a good target for gut inflammation in humans given the issues on myelotoxicity and thus studies in humans currently focus on Jak1 and to a lesser extent on Tyk2. Second the mouse model used is a model of acute intestinal injury and does not mimick chronic inflammation in humans. Finally, the authors showed prevention studies but did not demonstrate the therapeutic value of their compound.

Response (the first round of review): We thank the reviewer for the positive assessment on our strategy used for macrocyclic drug candidate design. Regarding the three issues of the colitis model that the reviewer is concerned about, we will elaborate one by one in the following sections.

1. First, I think that Jak2 is certainly not considered as a good target for gut inflammation in humans given the issues on myelotoxicity and thus studies in humans currently focus on Jak1 and to a lesser extent on Tyk2. Response: The import roles of JAK2 in the pathogenesis of inflammatory bowel disease (IBD) have been reported by many studies. For example, the study of Song et al. using intestinal mucosal biopsies has shown that the mRNA and protein expression of JAK2 were significantly increased in Chinese patients with ulcerative colitis (UC) and Crohn's disease (CD), two subtypes of IBD. In the study conducted by AsadzadehAghdaei ea al., 246 patients with IBD were enrolled, and JAK2 mRNA upregulation was also observed⁶. These findings suggest the potential of JAK2 as a therapeutic target for IBD. Although JAK2 inhibitors

have not been reported in clinical research and marketed drugs related to IBD, the investigational research continues. For example, Rivera revealed the overall protective role for intestinal epithelial JAK2 in intestinal homeostasis⁷. Besides, many traditional Chinese herbs have been reported to ameliorate DSS-induced colitis in mice via suppressing JAK2/STAT3 signaling. The exclusion of JAK2 from IBD treatment is mainly due to its engagement in signaling of hematopoietic cytokines, which may cause the undesirable side effect of myelosuppression. Ruxolitinib is a selective JAK2/JAK1 inhibitor, with IC₅₀ values of 2.8 and 3.3 nM, respectively. Ruxolitinib has been approved for treatment of myelofibrosis and polycythemia vera (PV), but it carries the undesirable side effect of myelosuppression. Swei and coworkers reported the successful use of Ruxolitinib to treat a patient with concomitant UC and PV¹⁰. The specific case sheds light on alternative therapies for UC treatment from a special perspective and underscores the importance to explore the broad efficacy and safety of selective JAK inhibitors. In summary, continued efforts are needed to clarify the potential therapeutic efficacy of JAK2 inhibitors in IBD. We hope that our preliminary exploration can play a role in attracting new ideas.

REPLY TO AUTHORS: Sorry but this is not convincing. Simply citing some papers on JAK2 expression does not implicate that this is a good target. Nobody in the IBD field would consider JAK2 as a good target (in contrast to JAK1). The authors have not addressed this point.

Response: At present, targeting JAK2 for the treatment of IBD JAK2 is controversial due to its potential side effects of myelosuppression. In this study, we mainly focus on the development of the deep learning-based macrocyclization method, and the present wet experimental validation is preliminary works. Therefore, in the revised manuscript, we highlighted the limitation of targeting JAK2 for IBD treatment in the last paragraph of the “*In vivo* activities of compound 3” section.

2. Second the mouse model used is a model of acute intestinal injury and does not mimick chronic inflammation in humans.

Response (the first round of review): Human IBD is a chronic, relapsing inflammatory disorder of the gastrointestinal tract, its etiology and pathogenesis are complicated and still uncertain. Due to its rapidity, simplicity, and reproducibility, the dextran sulfate sodium (DSS)-induced colitis model is widely used in IBD research as a preclinical model. Although there are differences between the acute DSS colitis model in our study and human IBD, they share many clinical and pathological features, such as bloody stools, weight loss, diarrhea, and inflammatory cells infiltration. The acute colitis model has been used for studying the pathogenesis of IBD as well as evaluating the efficacy of many drug candidates including those involved in JAK/STAT signaling.

REPLY TO AUTHORS: Sorry but this is not convincing. The authors have done nothing to address my point.

Response: The DSS-induced acute colitis model is commonly used to assess the efficacy of compounds in the initial study. In the revised manuscript, we highlighted the importance and necessity of the chronic IBD animal models to better mimic the chronic pathological conditions in humans.

3. Finally, the authors showed prevention studies but did not demonstrate the therapeutic value of their compound.

Response (the first round of review): As described in the In vivo Efficacy Study section of the Methods part of the manuscript, Male BALB/c mice were given 3.5% DSS water daily for 7 days to induce colitis, and the tested compounds were administered from day 8. As shown in Fig. 7, compound 3 could alleviate the decrease of body weight, reduce the disease activity index (DAI) score, and improve the colonic histological changes, demonstrating the therapeutic value of our compound.

REPLY TO AUTHORS: Sorry but this is not convincing. Data in chronic colitis models are needed.

Response: Thanks to the reviewer's suggestion. The issues related to the chronic colitis models will be scheduled in the follow-up work, in order to clarify the efficacy and safety of our macrocyclic compounds.